# Constraining sector-specific $CO_2$ and $CH_4$ emissions in the United States

Scot M. Miller[1] and Anna M. Michalak[1]

[1]Department of Global Ecology, Carnegie Institution for Science, Stanford, CA, USA

*Correspondence to:* Scot M. Miller (scot.m.miller@gmail.com)

**Abstract.** This review paper explores recent efforts to estimate state- and national-scale carbon dioxide ($CO_2$) and methane ($CH_4$) emissions from individual anthropogenic source sectors in the United States. Nearly all state and national climate change regulations in the US target specific source sectors, and detailed monitoring of individual sectors presents a greater challenge than monitoring total emissions. We particularly focus on opportunities to synthesize disparate types of information

5   on emissions, including emissions inventory data and atmospheric greenhouse gas data.

We find that inventory estimates of sector-specific $CO_2$ emissions are sufficiently accurate for policy evaluation at the national scale but that uncertainties increase at state and local levels. $CH_4$ emissions inventories are highly uncertain for all source sectors at all spatial scales, in part because of the complex, spatially-variable relationships between economic activity and $CH_4$ emissions. In contrast to inventory estimates, top-down estimates use measurements of atmospheric mixing ratios

10   to infer emissions at the surface; thus far, these efforts have had some success identifying urban $CO_2$ emissions and have successfully identified sector-specific $CH_4$ emissions in several opportunistic cases. We also describe a number of forward-looking opportunities that would aid efforts to estimate sector-specific emissions: fully combine existing top-down datasets, expand intensive aircraft measurement campaigns and measurements of secondary tracers, and improve the economic and demographic data (e.g., activity data) that drive emissions inventories. These steps would better synthesize inventory and top-

15   down data to support sector-specific emissions reduction policies.

# 1 Introduction

Government regulations of greenhouse gas (GHG) emissions have evolved rapidly in the past five years, particularly in the United States. The US pledged to decrease its GHG emissions by 26–28% relative to 2005 levels by 2025 as part of the Paris Agreement negotiated at COP21 (UNFCC). In parallel with this agreement, the US Environmental Protection Agency

(EPA) has finalized $CO_2$ and $CH_4$ emissions regulations for numerous source sectors under the White House Climate Action Plan (Executive Office of the President, 2013). Several US states have also taken aggressive action on emissions, including Massachusetts (Massachusetts Executive Office of Energy and Environmental Affairs, 2015) and California (Air Resources Board, 2014), among others.

    These policy actions require that scientists and government agencies quantify regional- and national-scale GHG emissions

from specific source sectors. In this paper, we define a source sector as the total emissions from an industry, such as $CO_2$ from power plants, $CH_4$ from the oil and natural gas industries, or $CH_4$ emissions from landfills. This review paper focuses on existing and evolving capabilities for the United States. The US has far greater resources to estimate emissions relative to many developing countries. Furthermore, GHG emissions regulations in the US are nascent relative to regulations in Europe (e.g., Prahl and Hofman, 2014), and the monitoring strategies discussed in this review could be developed in parallel with new

regulations.

    Many national emissions regulations in the US target this sector level. For example, the US Clean Power Plan mandates a 32% decrease in power sector $CO_2$ emissions by 2030 relative to 2005 levels (In February, 2016, the Supreme Court stayed implementation pending a final court ruling. The new presidential administration that assumed office in January 2017 has announced its intention to discard the plan.) (US EPA, 2015). The EPA and National Highway Traffic Safety Administration

have also extended and strengthened $CO_2$ emissions standards for cars and light trucks through 2025 (US EPA Office of Transportation and Air Quality, 2012). In addition to these measures, EPA has set several sector-specific $CH_4$ emissions targets. In May of 2016, EPA issued a rule that will decrease $CH_4$ emissions from oil and gas operations by 40–45% relative to 2012 levels by 2025 (US EPA, 2016b). In August of 2014, the US EPA, US Department of Agriculture (USDA), and US Department of Energy (DOE) released the *Biogas opportunities roadmap* targeting voluntary reduction strategies for agriculture (USDA

et al., 2014). Last but not least, EPA announced regulations for $CH_4$ emissions from landfills in July 2016 (EPA, 2016b). It is important to note that a number of these national policies are implemented at the state level. For example, each state has a different emissions reduction target under the Clean Power Plan, and each state can decide how to meet and monitor progress toward that target (US EPA, 2015).

    We examine sector-specific GHG estimates with an eye toward combining or assimilating multiple data streams. This review

article is part of a special issue of the European Geophysical Union (EGU) journals that focuses on data assimilation and the use of multiple data streams to understand the carbon cycle. In this context, we explore opportunities to creatively synthesize both bottom-up emissions inventories and top-down atmospheric inverse modeling. Most government agencies estimate emissions using bottom-up inventories: quantify total emissions by estimating the total amount of some activity and the average emissions per unit of activity. Other efforts utilize top-down atmospheric inverse modeling: measure atmospheric GHG mixing ratios and

use those measurements to infer the level and distribution of emissions at the Earth's surface. In the future, scientists and government agencies will likely need to combine these approaches to robustly estimate sector-specific emissions – frameworks that can synergistically leverage the information content of bottom-up datasets and top-down strategies using atmospheric GHG data. This review paper focuses on these opportunities.

These frameworks will need to address two key tasks: estimate the total quantity of GHG emissions from each source type and detect changes or trends in emissions from that source type. From the standpoint of inverse modeling, the former problem is more challenging than estimating total emissions and requires separating the space-time patterns of one emissions source from the patterns of other sources. In the latter case, we not only need to estimate a trend in total emissions but also need to attribute this trend to trends in specific source sectors. This challenge is complicated by changes in technology and changes

in the spatial or temporal distribution of individual source sectors. For example, hydraulic fracturing and horizontal drilling became widely used in the past decade (US Energy Information Administration, 2015). These operations utilize new equipment and operational practices, and the spatial distribution of drilling across the United States has changed during that time; these emissions are literally a 'moving target.'

These challenges are further complicated by GHG fluxes from the biosphere, particularly in the case of $CO_2$. Biospheric

and fossil fuel sources will be important to disaggregate from one another for sound policy evaluation These sources are often co-located and trends in one could be mistaken for trends in the other. In addition, future changes in biospheric $CO_2$ and $CH_4$ sources may be natural or human-caused (e.g., land use change, emissions induced by climate change, biological and/or geological carbon sequestration). Disentangling these natural and human causes will be challenging. Note that GHG fluxes from the biosphere and biological/geological carbon sequestration are beyond the scope of this review.

In this article, we explore the challenge of estimating sector-specific emissions from several perspectives. First, we discuss bottom-up inventory efforts. We then explore top-down strategies to estimate sector-specific emissions and the atmospheric datasets available to make both bottom-up and top-down estimates. Next, we highlight several new or novel approaches for estimating sector-specific emissions, and lastly, we close the review with a synthesis discussion of forward-looking opportunities for combining bottom-up and top-down strategies.

## 2   Bottom-up data

Bottom-up efforts typically use an accounting-type approach to estimate sector-specific emissions. The first step usually involves collecting activity data: a map or database of economic activity or behavior that leads to emissions. Examples include the amount of coal burned by power plants, the number of passenger cars and miles travelled, and the number of cows by location. A second step entails estimating a set of emissions factors (EFs) for each activity. EFs could include the $CO_2$ emis-

sions per kg of coal burned or the average $CO_2$ emissions per mile travelled by passenger cars. The product of these two numbers provides a bottom-up estimate of emissions for a given source sector. State and national governments in the US use this strategy to construct official emissions estimates (e.g., California Air Resources Board, 2015; EPA, 2016a). A number of academic and government efforts have produced bottom-up $CO_2$ and $CH_4$ emissions estimates at local/regional (e.g., Gately

et al., 2013; Jeong et al., 2014; Lyon et al., 2015; California Air Resources Board, 2015), national (e.g., Petron et al., 2008; Gurney et al., 2009; Gately et al., 2015; US EPA, 2013; Environment and Climate Change Canada, 2016; Maasakkers et al., 2016), and global scales (e.g., Rayner et al., 2010; Andres et al., 2011; Oda and Maksyutov, 2011; Olivier et al., 2014; European Commission, Joint Research Centre (JRC)/Netherlands Environmental Assessment Agency (PBL), 2016). In this section, we primarily discuss bottom-up data with an eye toward how this information can be combined with top-down strategies.

## 2.1 A prototypical example

We describe EPA's estimate of $CO_2$ emissions from coal-fired power plants as a prototypical example of how government agencies construct bottom-up inventory estimates. EPA describes the procedure that it uses to estimate $CO_2$ emissions in compliance with 2006 IPCC guidelines (US EPA, 2016a): first, the agency estimates activity data – coal use by source sector. EPA uses retail statistics from the electricity sector to estimate total consumption by each type of end user (e.g., residential, commercial, etc.). Second, EPA adjusts this activity data to account for non-combustion uses, double-counted emissions, and fuel exports/imports. For example, a coal gasification plant in North Dakota produces synthetic natural gas; this fuel is added to natural gas activity data and subtracted from the coal activity data. According to EPA, "Because this energy of the synthetic natural gas is already accounted for as natural gas combustion, this amount of energy is deducted from the industrial coal consumption statistics to avoid double counting" (US EPA, 2016b, p. A-31). Third, EPA estimates the carbon content of the coal. EPA uses Energy Information Administration (EIA) estimates of carbon content by coal rank and state of origin (Hong and Slatick, 1994). EPA then computes the weighted average carbon content of coal by state of origin and estimates the end use of coal produced in each state (e.g., electricity, industry, etc.). The agency uses this procedure to estimate the average carbon content (and EF) for each end use sector in the United States (US EPA, 2016a).

IPCC guidelines also require a reference approach: an additional verification or consistency check against fuel production, imports, and exports (EPA, 2016a). The new draft inventory then goes through expert review undertaken by a panel of technical experts. EPA revises its inventory estimate based upon this review and distributes the subsequent draft for public comment. At the conclusion of that process, EPA issues its finalized inventory estimate.

The approach outlined above is similar to many government inventories. More recently, a number of academic efforts have developed very different approaches that leverage novel data streams (e.g., satellite images lights at night) or that use gridded activity data, and these efforts are described in detail in the next section.

## 2.2 Recent bottom-up efforts

In the past ten years, inventory efforts have moved from coarse estimates that rely heavily on proxy activity data to spatially-resolved estimates that use specific activity data and EFs that are tailored to the heterogeneities in each emissions source.

A number of recent $CO_2$ inventory efforts have incorporated more comprehensive activity data or detailed EFs than previously available. At the regional scale, Gurney et al. (2012) and Gately et al. (2013) developed on-road $CO_2$ emissions estimates for Indianapolis and Massachusetts, respectively. Emissions in the latter study are within 8.5% of Federal Highway Administration fuel consumption statistics but differ from the commonly-used, global-scale EDGAR inventory (Olivier et al., 2014;

European Commission, Joint Research Centre (JRC)/Netherlands Environmental Assessment Agency (PBL), 2016) by 22.8%. The authors explain that many global-scale efforts use road density as a proxy for vehicle emissions but argue that the relationship between road density and emissions is not constant. Two subsequent studies (McDonald et al., 2014; Gately et al., 2015) estimate on-road $CO_2$ emissions for the entire United States at spatial resolutions down to 1 km$^2$. McDonald et al.'s
2014 emissions estimates differ from EDGAR by 20-80% at the municipal level, though the two inventories produce nearly identical national totals.

At the national scale, the VULCAN inventory (Gurney et al., 2009) is the most comprehensive academic effort to date. The inventory includes $CO_2$ emissions by sector at high spatial and temporal resolutions – 10km × 10km and sub-daily for the year 2002. Furthermore, the inventory uses more detailed activity data than government efforts. For example, the inventory identifies
emissions from individual point sources, a contrast to EPA's estimate which reports only county-level point source totals. At the global scale, the EDGAR anthropogenic emissions inventory (available for 1970–2010) has moved from a $1° × 1°$ lat/lon resolution to $0.1° × 0.1°$ (Olivier et al., 2014; European Commission, Joint Research Centre (JRC)/Netherlands Environmental Assessment Agency (PBL), 2016). In a separate effort, Andres et al. (2011) estimated $CO_2$ emissions for 80 countries for the years 1950–2006 with a particular focus on estimating the seasonal cycle of $CO_2$ emissions.
A number of studies have also incorporated more detailed activity data and EFs to estimate anthropogenic $CH_4$ emissions at both regional and national scales. At the regional scale, Jeong et al. (2014) and Lyon et al. (2015) estimated oil and gas $CH_4$ emissions from California for 2010 and the Barnett Shale region for 2013, respectively. Both studies find emissions that greatly exceed EPA's estimates. A relatively small fraction of emitters account for the majority of oil and gas emissions, and Lyon et al. (2015) argue that rigorous EFs capture this skewed distribution more effectively than those used by EPA. In addition to
these oil and gas inventories, Owen and Silver (2015) compiled field studies of $CH_4$ emissions from agriculture (e.g., cattle, sheep, and manure management). The authors explain that current emissions inventories use EFs from lab-based experiments, not field observations. These field observations imply much higher EFs that result in larger emissions more in line with existing top-down estimates. At the national scale, Maasakkers et al. (2016) created a gridded version of EPA's $CH_4$ inventory (0.1 × 0.1 lat-lon, monthly resolution for 2012). Maasakkers et al. (2016) point out that the spatial distribution of their estimate is
different from EDGAR, particularly for the oil and gas industries. Oil and gas emissions in EDGAR correlate with population density while emissions in Maasakkers et al. (2016) are concentrated in drilling basins.

A number of additional studies also employ novel inventory methodology or novel proxy datasets. For example, Oda and Maksyutov (2011) developed ODIAC (Open source Data Inventory of Anthropogenic $CO_2$ emission), a global, gridded $CO_2$ inventory constructed using a database of $CO_2$ point sources and satellite images of lights at night. Rayner et al. (2010) and
Asefi-Najafabady et al. (2014) developed a data assimilation framework known as FFDAS (Fossil Fuel Data Assimilation System). The authors used datasets like population density, carbon intensity of energy, and satellite images of lights at night, and they reported national emissions totals. Davis and Caldeira (2010) used a very different approach from any of the above studies. The authors built a $CO_2$ inventory based upon economic imports and exports and explored the idea of carbon 'leakage', the carbon emitted by one country to manufacture products that are then imported by another country. These studies do not

provide emissions estimates for each individual source sector, but ODIAC and FFDAS do incorporate novel datasets to separate out point sources (e.g., power plants) from non-point emissions.

EPA's GHG Reporting Program (GHGRP) represents an important advancement in government inventory efforts. EPA announced the GHGRP in 2009 and emissions reporting began in 2010 (US EPA, 2013). The GHGRP requires all entities that
emit over 25,000 metric tons of $CO_2$ equivalents to report their emissions to a national registry (US EPA, 2013). This reporting threshold is equivalent to the GHG emissions of 3,439 homes or 5,263 cars (EPA, 2015). The agricultural sector is excluded from this threshold and is not required to report its emissions. Despite these omissions, EPA estimates that 85–90% of US GHG emissions are covered under the GHGRP. Other recent studies, however, argue that the GHGRP is less complete than estimated by EPA for two reasons (e.g., Kort et al., 2014; Karion et al., 2015; Lan et al., 2015; Lavoie et al., 2015; Lyon et al., 2015;
Mitchell et al., 2015; Subramanian et al., 2015; Zimmerle et al., 2015). First, the emissions that are excluded from the GHGRP are sometimes larger than estimated by EPA, and second, the EFs used in the GHGRP are smaller than actual emissions from some source sectors like oil and natural gas.

## 2.3   Recent, direct measurements that support bottom-up efforts

Inventory development requires two different types of data: activity data and data that can be used to develop EFs. Activity
data can come from economic, census, and remote sensing datasets, among other possible data sources. These datasets differ from those used to develop EFs. The IPCC provides a database of EF estimates but encourages countries to take measurements of emitters or emitting processes to develop tailored, country-specific EFs (Goodwin et al., 2006). A number of observation strategies can directly support the development and evaluation of country-specific EFs. We discuss a number of recent efforts here as well as the advantages and challenges of using these datasets.

One observation strategy is to measure GHG mixing ratios near an emitter or a group of emitters. These observations, by factor of their targeted spatial scale, can be directly used to evaluate a single source type and develop corresponding EFs. For example, a number of studies report on direct GHG measurements from individual facilities. These include direct stack measurements of power plant $CO_2$ emissions (e.g., Teichert et al., 2003) and numerous recent studies of $CH_4$ emissions from oil and gas operations: measurements of emissions from pneumatic controllers (Allen et al., 2015), compressor stations
(Subramanian et al., 2015), transmissions and storage systems (Zimmerle et al., 2015), and abandoned wells (Kang et al., 2014). In addition, several site-level studies target agricultural emissions. Kebreab et al. (2008) and Sejian et al. (2010) review several measurement strategies, and Owen and Silver (2015) specifically review field studies of $CH_4$ emissions from manure.

On-road measurements provide a picture of emissions that is one spatial scale larger than direct facility observations. This strategy usually entails measuring trace gas mixing ratios from a ground-based vehicle either on public roads (e.g., Maness
et al., 2015) or private roads in partnership with the facility owner (e.g., Roscioli et al., 2015). Existing studies often target oil and gas facilities (e.g., Roscioli et al., 2015; Brantley et al., 2014; Jackson et al., 2014; Lan et al., 2015; Mitchell et al., 2015; Subramanian et al., 2015) and mobile $CO_2$ emissions (e.g., Brondfield et al., 2012; Maness et al., 2015). In the case of oil and gas emissions, Brantley et al. (2014) explain that mobile measurements capture an integrated plume that includes all leaks from a given facility but rarely indicate which components caused those leaks.

The use of facility-level and on-road observations entails a number of challenges. For example, facility-level observations provide the most insight into detailed emissions processes from specific source sectors but can miss emissions events or processes. Observations of oil and gas facilities provide a prime example; scientists may not know about some leaks and therefore may not measure them, other leaks may be in inaccessible locations (e.g., Subramanian et al., 2015), and the largest leaks often come from ephemeral equipment failures at a small number of facilities that are difficult to identify (e.g., Brantley et al., 2014; Allen, 2014; Allen et al., 2015). Cost also limits facility-level, continuous emissions monitoring; it is typically only used for large point sources like power plants (National Research Council, 2010).

These observation strategies also require extrapolation to produce state or national-scale EF estimates. The relationship between activity data and emissions can be complex and spatially variable, making it difficult to extrapolate facility or on-road measurements. For example, $CH_4$ emissions from oil and gas are likely dominated by a small number of malfunctioning facilities. As a result, it is difficult to develop robust, national-scale EFs from a modestly-sized sample of facilities (Allen, 2014). Furthermore, Brantley et al. (2014) explain that these leaks do not correlate with production and can vary greatly in time. Different oil and gas drilling basins also have different overall leakage rates – from 0.3% in Pennsylvania's Marcellus shale region to 8.9% in Utah's Uintah basin (e.g., Karion et al., 2013; Petron et al., 2014; Karion et al., 2015; Peischl et al., 2015). These factors make it challenging to create consistent, generalizable EFs that can translate activity data into emissions.

These considerations also apply to other source sectors beyond the oil and gas industries. For example, grazing and manure management practices differ by region, and manure and landfill $CH_4$ emissions also differ by climate (EPA, 2016a, ch. 5), all of which make extrapolation more challenging.

## 2.4 Impact of recent advances

Inventory estimates of sector-specific $CO_2$ emissions from the US are likely relatively accurate at national-scale but have substantial uncertainties at the local and state levels. Ackerman and Sundquist (2008), for example, compared smokestack versus fuel-based $CO_2$ estimates for US power plants and found a mean absolute difference of 16.6% but only a 1.4% total difference at the national scale. Furthermore, Gately et al. (2015) found biases of 100% or more at the urban scale in $CO_2$ emissions estimates for mobile sources. However, they estimated a US national total that was broadly consistent with other inventories like VULCAN.

By contrast, sector-specific $CH_4$ emissions are more challenging to estimate and existing inventories for the US are highly uncertain at state and national scales. For example, several top-down studies indicate that the California state inventory is likely too low by a factor of 1.2 to 1.9 (Jeong et al., 2013, 2016; Wecht et al., 2014b), and several top-down studies estimate emissions for oil and gas drilling regions of Utah and Colorado that are up to three times bottom-up estimates (e.g. Karion et al., 2013; Petron et al., 2014). Overall, total US $CH_4$ emissions are likely ∼50% larger than estimated by EDGAR or US EPA (Miller et al., 2013; Wecht et al., 2014a; Turner et al., 2015). Fig. 1 compares several inventory estimates of sector-specific $CO_2$ and $CH_4$ emissions. Existing $CO_2$ inventory estimates are broadly consistent while $CH_4$ estimates vary between inventories and among inventory versions.

$CH_4$ inventories are so uncertain, in part, because of the complexity of many anthropogenic $CH_4$ source sectors. For example, emissions factors for oil and gas operations are difficult to estimate because a small number of emitters often account for a large fraction of emissions (e.g., Allen, 2014; Brantley et al., 2014; Allen et al., 2015; Lan et al., 2015; Mitchell et al., 2015) and because there are so many points along the natural gas production, processing, transmission, and distribution cycle that leak methane (e.g., Kang et al., 2014; Allen et al., 2015; McKain et al., 2015; Subramanian et al., 2015; Zimmerle et al., 2015).

Much of the uncertainty in $CH_4$ inventories stems from difficulties developing accurate EFs. Brandt et al. (2014) writes, "... measurements for generating emission factors are expensive, which limits sample sizes and representativeness. Many EPA EFs have wide uncertainty bounds. And there are reasons to suspect sampling bias in EFs, as sampling has occurred at self-selected cooperating facilities." For example, EPA's EFs for natural gas pipelines are based on a limited number of samples from a 1996 EPA and Gas Research Institute study; these EFs have uncertainties of $\pm65\%$ (Beusse et al., 2014). Beyond the oil and gas industry, Owen and Silver (2015) also argue that many EFs for agriculture too low. These estimates are based upon a small number of pilot or lab experiments that were not explicitly designed for GHG inventory development.

## 3 Top-down, inverse modeling strategies

In this section, we discuss inverse modeling strategies – strategies that leverage observations of atmospheric GHG mixing ratios to infer emissions at the Earth's surface. We specifically focus on strategies that attempt to parse the contribution of specific source sectors. The first part of this discussion (Sects. 3.1 – 3.2) focuses on efforts at local, urban, and regional scales. These studies do not provide direct state- or national-level estimates but could be combined or extrapolated to quantify emissions at larger spatial scales. Many studies in this category target source sectors that do not overlap spatially, at least at the spatial scale of interest. The second part of this discussion (Sects. 3.3 – 3.4) explores inverse modeling efforts that directly estimate sector-specific emissions at the state and national levels. These efforts use observation networks that are sensitive to emissions across broad geographic regions. These efforts must also devise strategies to disentangle emissions from multiple, spatially overlapping source sectors.

### 3.1 Local-scale inverse modeling

Local-scale inverse modeling can best attribute emissions when the study region has a single, dominant source type. An estimate of total emissions for the region thus provides insight into the source sector of interest.

Studies that fall within this category often employ one of a few different strategies to estimate emissions. For example, many efforts use a simple box-modeling approach to estimate emissions (e.g., Turnbull et al., 2011; Karion et al., 2013; Caulton et al., 2014; Karion et al., 2015; Schneising et al., 2014; Cambaliza et al., 2015; Peischl et al., 2015) while others use an atmospheric transport model to relate GHG observations to emissions (e.g., McKain et al., 2012, 2015). Studies that use the former strategy typically estimate emissions in a few steps: first, make GHG measurements upwind and downwind of the region of interest. Second, use the difference between these measurements, the rate of flow through the "box" (i.e., wind speed adjusted by pressure), and the volume of the box (i.e., the area of the box and the mixing height of the atmosphere) to calculate

total emissions in the box. Most studies that use box modeling estimate a total flux for the region of interest, a number that is not spatially resolved.

Other studies in this category use a more involved approach: model atmospheric GHG mixing ratios using an emissions inventory and an atmospheric transport model. Subsequently, these studies scale the inventory using a single scaling factor ($\beta$) to better match modeled mixing ratios against measured mixing ratios:

$$y_k = \sum_{j=1}^{m_s \cdot m_t} h_{j,k}(x_j^a) + \epsilon_k \tag{1}$$

$$x_j^a = \beta x_j^b \tag{2}$$

In these equations, $y_k$ is an atmospheric GHG observation at a given time and location $k$. It is one of $n$ total observations ($k = 1...n$). The variable $x_j$ denotes the emissions from a model grid box $j$ at a specific location and time, and the function $h_{j,k}()$ is an atmospheric transport model that relates the surface emissions from grid box $j$ to observation $y_k$. The variable $m_s$ denotes the total number of model grid boxes in space, and $m_t$ denotes the number of time periods. In one study, this emissions estimate varied both spatially and temporally (McKain et al., 2012), and in another study, the emissions varied spatially but were constant in time ($m_t = 1$) (McKain et al., 2015). The superscripts $a$ and $b$ denote an emissions inventory and final emissions estimate, respectively. In addition, the variable $\epsilon_k$ denotes the cumulative error in the model and measurement (e.g., error in estimated transport, in the measurement, and in the estimated emissions, among other errors). The objective of this approach is to scale an inventory estimate using a single scaling factor ($\beta$) so that modeled atmospheric mixing ratios on the right hand side of Eq. 1 reproduces the $n$ observed atmospheric mixing ratios ($y_k$ where $k = 1...n$).

These local-scale efforts can target sources with very large emissions or very uncertain emissions. For example, many existing studies have targeted emissions from cities. Cities account for 70% of global fossil fuel $CO_2$ emissions, so insight into urban emissions provides insight into a large fraction of total anthropogenic GHG emissions (Energy Information Administration (EIA), 2016). Note that studies in this category generally do not discriminate among different urban source sectors but can provide insight into the contribution of urban $CO_2$ sources versus power plant $CO_2$ sources (which often occur well outside city limits). Existing efforts have estimated $CO_2$ emissions for Indianapolis, Indiana (Mays et al., 2009); Sacramento, California (Turnbull et al., 2011); and Salt Lake City, Utah (McKain et al., 2012) as well as $CH_4$ emissions from Boston, Massachusetts (McKain et al., 2015) and Indianapolis (Cambaliza et al., 2015). McKain et al. (2012) and McKain et al. (2015) used the approach in Eq. 1 while the other studies implemented box models.

Other studies in this category target oil and natural gas industry emissions. Existing studies have used aircraft observations to estimate $CH_4$ emissions from Utah's Uintah drilling basin (Karion et al., 2013), from southwest Pennsylvania (Caulton et al., 2014), from Colorado's Denver-Julesburg Basin (Petron et al., 2014), from the Barnett Shale in Texas (Karion et al., 2015; Lavoie et al., 2015), and from the Haynesville, Fayetteville, and Marcellus shale regions (in Texas, Arkansas, and Pennsylvania, respectively) (Peischl et al., 2015). In addition to these aircraft-based studies, one study used the SCIAMACHY instrument on the Envisat satellite to estimate $CH_4$ emissions from the Eagle Ford and Bakken shale regions in Texas and North

Dakota, respectively (Schneising et al., 2014). Several of these studies found leakage rates that greatly exceed EPA's estimated emissions factors (e.g., Karion et al., 2013; Petron et al., 2014; Schneising et al., 2014) while other studies estimated leakage rates that are comparable to EPA's numbers (e.g., Caulton et al., 2014; Peischl et al., 2015). Differences in drilling technology and practices from one basin to another may account for these contrasting results (e.g., Peischl et al., 2015).

These local-scale inverse modeling studies confer a number of advantages relative to other top-down strategies. These strategies capture emissions from all facilities in a given region, including those with anomalously high emissions. In the past, EPA has had difficulty designing facility-level measurements that adequately sample these anomalous emitters (Sect. 2.4). An additional advantage of these strategies is their ease of implementation relative to those discussed in subsequent sections (Sects. 3.3 – 3.4). Box modeling requires an estimate of air flow into and out of the box, but this approach does not

require a full atmospheric transport model. Furthermore, the strategies discussed in this section are not as computationally intensive as many of the state- and national-scale strategies discussed later in Sect. 3.3.

    These strategies also bring a number of challenges. Nearly all of the oil and gas studies listed above use data from a single measurement campaign and provide a temporal snapshot of emissions. Greenhouse gas emissions reduction policies make it necessary to monitor trends, a goal that requires sustained monitoring. In addition, a locality or region must have one dominant

source sector or have spatially (or temporally) non-overlapping source sectors in order to attribute emissions using this strategy (e.g., Hutyra et al., 2014; Peischl et al., 2015). For example, Peischl et al. (2015) estimated oil and gas emissions from drilling regions that also contain livestock, landfills, and wastewater treatment facilities, all of which produce $CH_4$ emissions. The authors subtracted an inventory estimate of these non-hydrocarbon $CH_4$ sources from their estimated emissions total, and they attributed the remaining emissions to oil and gas activities. The authors point out that these non oil and gas source sectors are

small contributors relative to oil and gas operations (8.5 – 19% of the $CH_4$ emissions total in each region), and uncertainties in these other source sectors would likely have a small impact on their oil and gas emissions estimate.

    Complex environmental conditions and the associated atmospheric transport errors can also pose a challenge for local-scale inverse modelings strategies, particularly for box models. A simple box modeling setup can be difficult to apply when atmospheric advection, vertical mixing, or upwind "clean air" measurements are highly heterogeneous across the box. For

example, Turnbull et al. (2011) report that their $CO_2$ budget for Sacramento, estimated using a box model, is uncertain by a factor of two due to uncertainties in estimated wind speed and upwind "clean air" mixing ratios. Furthermore, Karion et al. (2015) estimated $CH_4$ emissions for the Barnett Shale that varied from $4.4 \times 10^4$ to $10.9 \times 10^4$ kg hr$^{-1}$, depending on the flight. However, the authors explain that two of the eight flights occurred during non-ideal meteorological conditions, and the range of estimates narrowed to $6.1 \times 10^4$ to $8.8 \times 10^4$ kg hr$^{-1}$ when those flights are excluded from the analysis. Atmospheric transport

models can simulate more complex atmospheric transport patterns relative to box models but still have difficulty modeling local- or urban-scale phenomena, including small-scale turbulent eddies, air flow through street canyons, and vertical mixing in a human-built landscape (e.g. Nehrkorn et al., 2013). These modeling challenges also apply to the state- and national-scale strategies discussed in Sects. 3.3 – 3.4. New innovations in atmospheric monitoring and instrumentation may reduce some of these uncertainties. Cambaliza et al. (2014), for example, explain that LIDAR instruments can measure atmospheric mixing

height, and LIDAR deployment could therefore improve certain aspects of atmospheric modeling, particularly at local and

regional scales. In addition, several studies have developed high resolution meteorological simulations, in part to better resolve atmospheric GHG transport in urban environments (e.g., McKain et al., 2012; Nehrkorn et al., 2013; McKain et al., 2015).

## 3.2 Observations that support local-scale inverse modeling

Many recent, local-scale observation efforts have focused on urban monitoring and on oil and gas basins. Existing urban, atmospheric measurement networks include Salt Lake City, Utah (McKain et al., 2012); Los Angeles, California (Duren, 2016); Oakland, California (Cohen, 2016), the Bay Area Air Quality Management District (Fairley and Fischer, 2015), and Indianapolis (Mays et al., 2009; Cambaliza et al., 2015; Lauvaux et al., 2016). Recent local-scale aircraft campaigns include the INFLUX campaign focused on the Indianapolis metro region (Cambaliza et al., 2015), the SENEX and SOGNEX campaigns focused on multiple oil and gas drilling basins (Peischl et al., 2015; NOAA Chemical Sciences Division, 2016), and the Barnett Coordinated Campaign (Smith et al., 2015; Karion et al., 2015) (Fig. 2). In addition to these urban and oil and gas studies, Lindenmaier et al. (2014) used ground-based, $CO_2$ column observations to identify emissions from a large coal-fired power plant in the Four Corners region of the western US.

The observational strategies described above are relatively diverse. These efforts include a combination of aircraft and stationary sites (e.g., telecommunications towers or building rooftops). Some of these campaigns provide a one or two day snapshot in time (e.g, most oil and gas studies) while other campaigns involve sustained measurements over a year or more (e.g., urban observation networks like LA Megacities and the Indianapolis INFLUX project).

## 3.3 State- and national-scale inverse modeling

The top-down strategies discussed in this section provide sector-specific GHG emissions estimates across larger regions, regions that typically have several overlapping source sectors. Furthermore, these strategies make spatially variable adjustments to existing inventories, unlike the strategies outlined in Sect. 3.1. The three strategies discussed in this section use both GHG observations and inventories to attribute sector-specific emissions. Each approach, however, uses a different mix; the first approach relies most heavily on existing inventories while the last relies most on GHG observations.

Overall, these strategies have been relatively successful at attributing $CH_4$ emissions, but promising strategies for $CO_2$ are nascent. Biospheric $CO_2$ fluxes are large relative to anthropogenic $CO_2$ emissions at diel to monthly time scales, particularly during the growing season, and the spatiotemporal distribution of these fluxes is highly uncertain (e.g., Huntzinger et al., 2012). These factors have limited the success of $CO_2$-focused efforts.

The first strategy discussed here scales the individual source sectors in a bottom-up inventory. This setup is often similar to a multiple linear regression:

$$x_j^a = \sum_{i=1}^{p} \beta_i x_{i,j}^b \tag{3}$$

where $i$ denotes an individual source sector from a bottom-up inventory, and $p$ indicates the total number of source sectors in the inverse model. The observational constraint ($y_k$ where $k = 1...n$) in this approach is the same as in Eq. 1. This setup

also assumes that the initial emissions estimate ($x_{i,j}^b$ where $i = 1...p$ and $j = 1...m_s \cdot m_t$) is defined at each of $m_s$ spatial locations, at each of $m_t$ time periods, and for each of $p$ source sectors. In one study, this initial emissions estimate was spatially but not temporally resolved (e.g., $m_t = 1$) (Zhao et al., 2009), while in another study, it was resolved in both space and time (Jeong et al., 2013). The $p$ unknown scaling factors ($\beta_i$ where $i = 1...p$) adjust the magnitude of different source sectors in the

bottom-up inventory; these factors are estimated by the inverse model. As a result of this setup, the estimated emissions ($x_j^a$) will always be a linear combination of source-specific emissions patterns in an existing bottom-up inventory. Studies that use this approach often estimate the scaling factors ($\beta_i$) using Bayesian statistics; these frameworks can weigh uncertainty in the measurements ($y_k$) and in the atmospheric model ($h_{j,k}$) against uncertainty in the initial or prior guess for the scaling factors (typically unity) (e.g., Rayner et al., 2016).

To date, a handful of studies have leveraged this approach to attribute emissions of $CH_4$. For example, Zhao et al. (2009) and Jeong et al. (2013) used atmospheric measurements from tall towers to estimate emissions from individual source sectors in California. Both studies found higher $CH_4$ emissions from agriculture relative to the EDGAR emissions inventory.

This scaling factor approach brings several strengths and weaknesses. An advantage of this approach is that it not only provides an estimate of total emissions but also the contributions of individual source sectors. The approach can be relatively

easy to implement from a statistical perspective. With that said, one still needs to run an atmospheric transport model and must have an estimate of background or upwind, clean air mixing ratios.

A notable challenge of this strategy is that it requires accurate knowledge of the spatial distribution of each source sector. The estimated emissions will always be a linear combination of source-specific emissions patterns from an existing inventory, and errors in the spatial distribution of these inventories will propagate into errors in sector-specific attribution. Furthermore,

the atmospheric GHG observations ($y_k$ where $k = 1...n$) must be sensitive to differences in the space-time patterns among different source sectors. Worded differently, each of the $p$ source sectors must have differing spatiotemporal patterns, and each sector must explain substantial variability the observations ($y_k$). If the former condition does not hold, then some of the $p$ source sectors will be collinear; colinearity can lead to unphysical scaling factors ($\beta_i$ where $i = 1...p$) and unrealistically large uncertainty estimates (e.g., Zucchini, 2000). If the latter condition does not hold, then the scaling factors may be poorly

constrained by the data, resulting in uncertain or unrealistic sector-specific estimates. To account for these challenges, Jeong et al. (2013) only reported source-specific estimates when they obtained scaling factors that were statistically significantly different from zero.

A second common inverse modeling strategy scales an emissions inventory at the model grid level to better reproduce the atmospheric observations ($y_k$ where $k = 1...n$). All of the strategies discussed previously scale the spatial patterns in an

existing inventory. By contrast, this strategy scales the emissions level at each location in the model domain, and the resulting estimate ($x_j^a$ where $j = 1...m_s \cdot m_t$) can have spatial patterns that are different from any inventory ($x_j^b$). These estimates have the following general form:

$$x_j^a = \beta_j x_j^b \tag{4}$$

Note that $x_j^b$ and $x_j^a$ are the total emissions from model grid box $j$, not the emissions by sector. Hence, the scaling factors ($\beta_j$ where $j = 1...m_s \cdot m_t$) adjust total emissions, and all of the $m_s \cdot m_t$ factors are typically estimated simultaneously. Several studies estimate scaling factors that vary spatially but are the same at each time step (e.g., Wecht et al., 2014a, b; Turner et al., 2015). One study allows the scaling factors to vary in both space and time (Jeong et al., 2016). This approach is also Bayesian in nature; the modeler sets an initial guess for the scaling factors (typically unity) and an uncertainty in that initial guess; this information guides the estimate for the scaling factors, particularly when these factors are under-constrained by the available observations ($y_k$ where $k = 1...n$) (e.g., Rayner et al., 2016).

This strategy does not support attribution in and of itself, but several studies have adapted this approach for source attribution. These studies attribute emissions in each model grid cell using the attribution in a bottom-up inventory. For example, let's say that an inventory estimates that 60% of the emissions in a given grid cell are from oil and gas and 40% are from cattle and manure. The inverse modeling estimate will attribute emissions in that grid box in the same proportion:

$$x_{i,j}^a = \beta_j x_{i,j}^b \tag{5}$$

All variables in this equation are as defined earlier. As a result of this setup, the total emissions in any one model grid box may differ from the inventory. However, the relative magnitude of the source sectors in any one grid box will be the same as in the bottom-up inventory.

Wecht et al. (2014b) and Jeong et al. (2016) leveraged this strategy to estimate $CH_4$ emissions for California using aircraft and tower-based observations, respectively. Like Zhao et al. (2009) and Jeong et al. (2013), they also found higher emissions from agriculture relative to EDGAR. Wecht et al. (2014a) and Turner et al. (2015) further applied this strategy to attribute emissions at continental scales; these studies used Envisat/SCIAMACHY and the GOSAT satellite, respectively, to estimate sector-specific $CH_4$ emissions across North America. Both studies estimated larger emissions from agriculture relative to the EPA and EDGAR inventories. Turner et al. (2015) estimated oil and gas emissions that are a factor of two larger than EDGAR while Wecht et al. (2014a) found that these emissions are broadly consistent with EDGAR.

This strategy has a number of advantages and weaknesses relative to other approaches. The strategy can be used to estimate emissions at grid scale, and the resulting emissions estimate will not be the a linear combination of existing inventory estimates. However, it assumes that the inventory has correctly estimated the relative magnitude of each emissions source in each model grid box. Errors in this relative magnitude will produce errors in the sector-specific attribution.

Third a number of studies have leveraged a strategy known as geostatistical inverse modeling (GIM) to estimate GHG fluxes generally (e.g., Michalak et al., 2004; Gourdji et al., 2008, 2012) and anthropogenic emissions specifically (Miller et al., 2013, 2016; Shiga et al., 2014; ASCENDS Ad Hoc Science Definition Team, 2015; Yadav et al., 2016). This approach attributes patterns in the emissions to individual anthropogenic source sectors when possible. However, it will leave emissions as unattributable when those emissions do not match the space-time patterns in any bottom-up inventory or when the information content of the atmospheric observations is insufficient for attribution:

$$x_j^a = \sum_{i=1}^{p} \beta_i x_{i,j}^b + \xi_j \tag{6}$$

The elements $x_{i,j}^b$ (where $i = 1...p$ and $j = 1...m_s \cdot m_t$) can be individual source sectors from a bottom-up inventory (similar to Eq. 3). The inverse model will then map the emissions onto those patterns to the extent possible. The inverse model will further add (or subtract) emissions at the model grid scale to better reproduce the atmospheric observations ($y_k$ where $k = 1...n$). These emissions are denoted by $\xi_j$ (where $j = 1...m_s \cdot m_t$), and a GIM typically labels the emissions in $\xi_j$ as unattributable.

Furthermore, existing studies allow $x_{i,j}^b$ and $\xi_j$ to vary both spatially and temporally with $j$, in contrast to the studies described earlier in this section. Note that existing GIM studies have fixed the coefficients ($\beta_i$) in both space and time. In reality, the relationship between $x_{i,j}^b$ and GHG emissions may vary spatially and temporally by grid box $j$. Two recent GIM studies have experimented with allowing the coefficients to vary by region or biome in the context of anthropogenic (Shiga et al., 2014) and biospheric (Fang and Michalak, 2015) fluxes.

Several studies have leveraged this strategy in the context of both anthropogenic $CH_4$ and $CO_2$ emissions. Miller et al. (2013) used a GIM and in situ atmospheric measurements to estimate sector-specific $CH_4$ emissions in the US; like Turner et al. (2015), they found higher emissions from the agriculture and oil and gas sectors relative to inventory estimates. Miller et al. (2016) also used this strategy to separate $CH_4$ emissions patterns due to wetlands from anthropogenic emissions and to evaluate bottom-up estimates of the former emissions category. Two studies (Shiga et al., 2014; ASCENDS Ad Hoc Science

Definition Team, 2015) implemented a GIM-based framework to identify anthropogenic $CO_2$ emission patterns using in situ and satellite $CO_2$ observations, respectively. They investigated whether the atmospheric signal resulting from anthropogenic $CO_2$ emissions could be reliably identified given the confounding signal from biospheric $CO_2$ fluxes. They found that in situ and remote sensing $CO_2$ networks could only identify anthropogenic emissions in a few regions during a few months of the year.

The GIM approach makes more conservative assumptions relative to other source attribution strategies discussed in this section. A GIM will only attribute emissions to patterns in a bottom-up inventory when that inventory matches patterns in the atmospheric GHG observations. In Miller et al. (2013), for example, the GIM mapped 60% of total US $CH_4$ emissions onto patterns in the EDGAR inventory and found that 40% of the total emissions were unattributable to the patterns in any bottom-up dataset. By contrast, the other approaches discussed above will attribute 100% of the emissions. In GIM studies

like Miller et al. (2013), the unattributable emissions indicate shortfalls in either the greenhouse gas observation network or available bottom-up data. In the former case, existing atmospheric observations do not provide enough information to reliably estimate sector-specific emissions patterns. For example, the information content of the atmospheric observations in Miller et al. (2013) was insufficient to uniquely constrain emissions from coal mining, and those emissions were included in $\xi_j$ instead of $\sum_{i=1}^{p} \beta_i x_{i,j}^b$. In the latter case, the unattributable emissions in $\xi_j$ indicate inaccuracies in the spatial distribution of

available inventory estimates. Existing inventories did not have well-developed activity data for the oil and gas industry, and the unattributable emissions in Miller et al. (2013) provide information about shortfalls in these activity datasets.

Yadav et al. (2016) modified the existing GIM framework to better isolate anthropogenic $CO_2$ emissions. The authors exploited differences in the spatiotemporal properties of biospheric versus fossil fuel fluxes to do this attribution. Specifically, the authors argued that the biospheric fluxes have smooth spatiotemporal patterns, and fossil fuels emissions do not have

smooth patterns. The authors then partitioned $\xi_j$ into two components (smooth and non-smooth) and attributed these emissions

to the biosphere and fossil fuels, respectively. The study examined emissions in January when biospheric fluxes are smaller than in other months.

In summary, this section discuss statistical innovations that help isolate individual emissions sources. In addition to these innovations, accurate models of atmospheric transport also play a crucial rule. A number of studies indicate the deleterious influence of transport errors. For example, Shiga et al. (2014) argue that atmospheric transport errors hinder the detection of fossil fuel emissions patterns across the United States. The authors also argue that biospheric fluxes mask fossil fuel patterns to a similar degree. Numerous additional studies examine the effects of transport errors on $CO_2$ modeling, though not in the context of fossil fuel emissions (e.g. Stephens et al., 2007; Liu et al., 2012; Miller et al., 2015).

Several efforts could reduce these transport modeling errors. Like urban-scale studies (Sect. 3.1), national inverse modeling studies have also begun moving toward high resolution meteorology simulations. These studies simulate atmospheric GHG transport at high resolution over the US and Canada and utilize coarser resolutions elsewhere to save on computational costs. For example, national-scale studies using the Weather Research and Forecasting (WRF) have modeled GHG transport at resolutions up to 8–10km (Nehrkorn et al., 2010; Gourdji et al., 2012; Miller et al., 2013), and studies using the GEOS-Chem model have simulated $CH_4$ transport at resolutions up to ~50km (e.g., Wecht et al., 2014a; Turner et al., 2015). In addition to these efforts, NASA's Atmospheric Carbon and Transport – America campaign (ACT–America, Fig. 2a) aims to diagnose and reduce atmospheric transport errors (NASA). The campaign includes new tower sites and five years of aircraft flights across the eastern US. Many flights will travel through frontal systems and extratropical cyclones to better characterize and evaluate atmospheric transport errors.

## 3.4 Observations that have been used to attribute emissions at state and national scales

The observations discussed in this section do not provide a direct constraint on an individual source sector but have been used by existing regional- and national-scale inverse modeling studies (Sect. 3.3) to support sector-specific attribution. These observations are typically distributed across a broad geographic region. They are therefore sensitive to emissions over a large area and can constrain larger regions, albeit with less detail than the local approaches discussed in Sect. 3.2.

Observations in this category include air samples collected atop telecommunications towers and from aircraft: the NOAA tall tower observation network (Andrews et al., 2014), regular NOAA aircraft monitoring (Sweeney et al., 2015), the Environment and Climate Change Canada tower monitoring network (Environment and Climate Change Canada, 2011), the California Greenhouse Gas Research Monitoring Network (e.g., Zhao et al., 2009; Jeong et al., 2012, 2013, 2016), and a privately-funded tower network operated by Earth Networks (Fig. 2). Most of the inverse modeling studies discussed in the previous section (Sect. 3.3) used these in situ observation networks to estimate sector-specific emissions (Zhao et al., 2009; Jeong et al., 2013; Miller et al., 2013; Shiga et al., 2014; ASCENDS Ad Hoc Science Definition Team, 2015; Jeong et al., 2016).

The current tower network is sensitive to emissions from some source sectors but not to others. Many of the NOAA tall towers and regular aircraft sites are in or near the Great Plains. As a result, the network has sensitivity to agricultural emissions and to several oil and gas basins but has little sensitivity to emissions from east coast population centers. Earth Networks, by contrast, has focused its efforts on the East Coast proximal to large population centers. The state of California has a dense

network of publicly-operated towers. By contrast to these regions, the network is sparse across the western US outside of California and northern Colorado. On one hand, the population in the regions is sparse and some emissions sectors are likely to be small (e.g., vehicle emissions). On the other hand, large resource extraction regions are beyond reach of the long term monitoring network, regions like the Powder River Basin coal mining region of Wyoming or the Bakken oil and gas basin in
Montana and North Dakota.

NOAA's regular aircraft monitoring network complements these tower-based sites. The flights measure GHG mixing ratios across a vertical atmospheric profile. These datasets can help evaluate vertical mixing and transport in atmospheric transport models, and observations from the middle and upper troposphere can be used to quantify background "clean air" concentrations, a necessity for the inverse modeling studies described in Sect. 3.3. A downside is that NOAA's aircraft profiles are usually
limited in frequency to one or two times per month, unlike towers which often have continuous observations. Scientists at NOAA have also invented a technology known as AirCore that can observe vertical atmospheric GHG profiles from a weather balloon (Karion et al., 2010). This technology could become a key component of the long term monitoring network in the future.

A number of intensive aircraft campaigns provide observations across entire state or multi-state regions (Fig. 2). These
include the 2010 CalNex campaign (Ryerson et al., 2013), the 2013 SEAC[4]RS campaign (Toon et al., 2016), and the ACT-America campaign (2015–2019) (NASA). Few studies have used these observations to attribute state-wide emissions. For example, Wecht et al. (2014b) used CalNex data to attribute state-wide $CH_4$ emissions from California.

Several satellites make total column observations of $CO_2$ and $CH_4$ (e.g., AIRS, TES, IASI, Envisat/SCIAMACHY, GOSAT, OCO-2, and GHGSat). Streets et al. (2013) describe a number of these satellites in detail, and Jacob et al. (2016) provide a
thorough overview of $CH_4$-observing satellites. Four of these satellites (Envisat/SCIAMACHY, GOSAT, OCO-2, and GHGSat) observe in the shortwave infrared. Relative to other satellites, these four are more sensitive to GHG mixing ratios in the lower troposphere and, hence, to emissions at the surface (e.g., Chevallier et al., 2005; Wecht et al., 2012). Only a handful of studies have used these datasets to attribute sector-specific emissions in the US, and these existing studies focus on $CH_4$, not $CO_2$ (e.g., Schneising et al., 2014; Wecht et al., 2014a, b; Alexe et al., 2015; Turner et al., 2015). For example, Turner et al. (2015)
used GOSAT observations to estimate sector-specific $CH_4$ emissions in North America and found results that were broadly consistent with emissions estimates derived from the US tall tower and aircraft monitoring network (Miller et al., 2013). Wecht et al. (2014b), however, explains that GOSAT observations are too sparse to constrain $CH_4$ emissions from California outside of the Los Angeles Basin.

## 4   Novel strategies that could be used for estimating sector-specific emissions

This section discusses two observational strategies that support top-down modeling efforts, strategies that show promise for estimating sector-specific emissions. First, we discuss the potential of upcoming and proposed satellite-based GHG observations. Next, we discuss the utility of 'secondary tracers.' These gases or isotopologues are co-emitted with GHGs and aid in sector-specific attribution.

## 4.1 New satellite-based GHG observations

Existing satellites could hold enormous potential for estimating fossil fuel emissions. For example, several studies indicate that Envisat/SCIAMACHY and GOSAT should be able to constrain $CO_2$ emissions from large cities or large industrial regions (e.g., Schneising et al., 2008; Kort et al., 2012; Schneising et al., 2013). Kort et al. (2012) further argues that GOSAT could detect a trend as small as 22% from Los Angeles. OCO-2 and GHGSat should be even more capable. OCO-2 observations have a smaller footprint and precision relative to GOSAT. As a result, the satellite should be able to constrain $CO_2$ from large power plants (National Research Council, 2010). The privately-funded GHGSat makes targeted observations over specific point sources with a smaller footprint than OCO-2 and therefore should be ideal for constraining large point sources (Kramer, 2017).

Other studies offer a more skeptical perspective on current satellite capabilities. Keppel-Aleks et al. (2013) argue that variations in total column $CO_2$ due to fossil fuel emissions are largely obscured by biospheric fluxes. Furthermore, Gavrilov and Timofeev (2015) found large biases ($4.7 \pm 2.6$ ppm) in GOSAT retrievals of $CO_2$. Future retrieval improvements could reduce these biases (e.g., Dils et al., 2014; Buchwitz et al., 2015). An additional challenge is that current satellites do not provide comprehensive global mapping and therefore are not well-suited for monitoring all urban areas and point sources (Fig. 2); Miller et al. (2007) point out that OCO-2 covers only 7–12% of Earth's land surface. Trend detection can also be challenging. Individual satellites have limited lifetimes, and different satellite datasets with unique error characteristics and biases can be difficult to compare.

Future satellites, both selected and proposed, offer a number of improvements over existing capabilities. Some, like GOSAT-2 (selected), have better precision relative to the existing generation of satellites (Matsunaga and et. al., 2016). Other future satellites have a wide swath (CarbonSat, proposed) or are geostationary (GeoCARB and GEO-CAPE; selected and proposed, respectively). They would generate higher density observations across the US relative to OCO-2 and GOSAT (Fishman et al., 2012; Polonsky et al., 2014; Bovensmann et al., 2015; Buchwitz et al., 2013; Bousserez et al., 2016; Pillai et al., 2016). LIDAR-based missions (e.g., MERLIN and ASCENDS; selected and proposed, respectively) measure in the absence of sunlight and through thin or scattered clouds (Kiemle et al., 2011; ASCENDS Ad Hoc Science Definition Team, 2015). As a result, these satellites would also generate dense observations relative to current satellites, particularly at high latitudes.

These future satellites should have sufficient precision and small footprints to constrain $CO_2$ emissions from power plants. They should also have better spatial coverage to monitor a greater number of emitters. For example, (Bovensmann et al., 2010) report that the proposed CarbonSat satellite should be able to constrain $CO_2$ emissions from a mid-sized power plant to within 12–36%. Other studies, by contrast, indicate that future missions like ASCENDS would have difficulty constraining regional-scale fossil fuel $CO_2$ emissions from the US (ASCENDS Ad Hoc Science Definition Team, 2015) and would have limited ability to detect continental-scale changes in emissions (Hammerling et al., 2015). In addition to $CO_2$, future $CH_4$ observations also show promise. For example, the TROMPOMI sensor is schedule to launch in 2017 and should be sufficient to constrain the largest 1% of grid cells in EPA's gridded $CH_4$ inventory (Maasakkers et al., 2016), equivalent to 30% of total national emissions (Jacob et al., 2016).

## 4.2   Secondary tracers

Secondary tracers are co-emitted with GHGs and are often emitted from only a small number of source sectors. These tracers make it possible to isolate and factor out at least a portion of natural fluxes or factor out emissions from source sectors that are not of primary interest. The top-down approaches discussed previously either require a limited geographic scope or accurate

activity data to effectively estimate sector-specific emissions. Secondary tracers could identify sector-specific emissions without these limitations (though secondary tracers present challenges of their own). Examples of secondary tracers include radiocarbon ($^{14}$C), ethane, $^{13}CO_2$, $^{13}CH_4$, and carbon monoxide (CO). We focus on radiocarbon and ethane because they hold particular promise.

### 4.2.1   Radiocarbon

Radiocarbon is produced by cosmic rays in the upper atmosphere and has a lifetime of approximately 5,730 years before decaying back to $^{12}$C (Bowman, 1990). Since the 1940s, nuclear bomb testing has elevated $^{14}$C within the atmosphere. $CO_2$ fluxes from the biosphere will mirror the isotopic composition of the atmosphere at the time that carbon was incorporated into the plant. $CO_2$ emissions from fossil fuels, by contrast, contain no $^{14}$C because fossil fuel reservoirs are far older than the decay lifetime of $^{14}$C, and these reservoirs have not interacted with atmospheric carbon during the intervening time period.

Several exploratory studies used radiocarbon to separate the atmospheric $CO_2$ signal from biogenic versus anthropogenic emissions. One study used radiocarbon measurements from the US East Coast to estimate the relative contribution of fossil fuel versus biogenic emissions (Miller et al., 2012). Another study reported on radiocarbon measurements in California (Riley et al., 2008). Graven et al. (2011) and LaFranchi et al. (2013) used radiocarbon observations from an aircraft and a tall tower, respectively, to estimate the contribution of anthropogenic and biogenic $CO_2$ emissions in Colorado. Beyond these studies,

radiocarbon measurements are not widely used in regional- or continental-scale inversions.

Radiocarbon has not been widely used, in part, because only a handful of atmospheric monitoring sites in the US report radiocarbon measurements. An expanded observation network shows enormous potential. NOAA and its partners currently measure radiocarbon in air samples from eight tall tower sites, three mountaintop sites, and four aircraft sites in the US. NOAA collects these samples up to three times per week at tall tower and mountaintop sites and collects up to two to three samples

every two weeks at aircraft sites. Basu et al. (2016) explain that there were 1639 total radiocarbon measurements between July 2009 and April 2011 (21 total months). By contrast, the National Research Council (2010) recommended that the US invest $15–20 million annually to collect 5000-10000 radiocarbon observations per year, but that goal has not yet come to fruition. Basu et al. (2016) argued that this level of investment would allow scientists to constrain US fossil fuel $CO_2$ emissions to within 1% per year and to within 5% per month.

Despite this promise, the use of atmospheric radiocarbon measurements also presents several challenges. One primary challenge is accounting for the disequilibrium effect (Bowman, 1990). The atmospheric abundance of $^{14}$C has changed in the past 75 years due to nuclear bomb testing. $CO_2$ from decomposing organic matter (heterotrophic respiration) will reflect $^{14}$C levels during the time that carbon was incorporated into plant tissue, not current atmospheric levels of $^{14}$C. Furthermore, the lifetime

of dissolved gases in the ocean is much longer than 75 years, so the isotopic signature of air-sea gas exchange will also lag the recent rise in atmospheric $^{14}C$. One must account for this mismatch or 'disequilibrium' when using radiocarbon measurements to partition between fossil fuel $CO_2$ and biospheric $CO_2$; biospheric (and ocean) fluxes will not necessarily match current atmospheric $^{14}C$ levels but rather reflect the levels of a past date. Atmospheric sampling upwind of anthropogenic sources could

be used to characterize the biospheric $^{14}C$ signature and would mitigate this concern.

### 4.2.2   Ethane

Methane is the primary component of natural gas, but natural gas also contains small quantitates of other alkanes, including ethane. These trace constituents are collectively referred to as natural gas liquids. Enhancements in atmospheric ethane mixing ratios indicate leaks from natural gas and oil infrastructure because these operations are a primary source of ethane to the

atmosphere (e.g., Rudolph, 1995). Other $CH_4$ emitters, including agriculture, landfills, and wetlands do not emit higher order alkanes in substantial amounts. For example, Peischl et al. (2013) estimated that natural gas leaks account for 90% of all ethane emissions in the Los Angeles metro region. If one has an estimate of ethane emissions and an estimate of the ethane content of natural gas, then one can estimate $CH_4$ emissions from oil and gas infrastructure. McKain et al. (2015), for example, measured $CH_4$ and ethane at several sites in Boston, and they used $CH_4$-ethane ratios reported from natural gas pipeline operators to

estimate the portion of Boston's $CH_4$ emissions that are due to natural gas leaks. Several other studies have similarly used ethane measurements to explore oil and gas industry emissions from Los Angeles (Wennberg et al., 2012), Dallas, Texas (Yacovitch et al., 2014), the Barnett shale region (Smith et al., 2015; Townsend-Small et al., 2015), and from global oil and gas operations (e.g., Simpson et al., 2012; Schwietzke et al., 2014).

The use of ethane for $CH_4$ source attribution brings several challenges. Until recently, atmospheric observations of ethane

were sparse. Research groups at UC-Irvine and NOAA have measured ethane in air samples from global background sites since 1984 and 2004, respectively (Simpson et al., 2012; Helmig et al., 2016). Each group collects samples at 40–45 sites at weekly to seasonal frequencies. Recently, NOAA has expanded its ethane measurements to its US tall tower and aircraft network. Instrumentation has also become more widely available with Aerodyne, Inc.'s ethane analyzer (Yacovitch et al., 2014).

The ethane content of natural gas can also vary by region and will change if natural gas liquids are removed at processing

facilities (Fig. 3). These variations complicate the task of inferring $CH_4$ emissions using ethane measurements. Smith et al. (2015), for example, found three distinct ethane signatures in different areas of the Barnett shale region. Townsend-Small et al. (2015) report that emissions operations in the Barnett ranged from 6% ethane at natural gas wells to 13% ethane at oil wells.

In summary, secondary tracers like ethane and radiocarbon allow scientists to leverage measurements networks with broad spatial coverage (like those in Sect. 3.4) to estimate specific source sectors. These measurements bypass, to some degree, the

need to rely on the spatial and temporal patterns in an inventory for source attribution and the need to have accurate activity data to support inverse modeling. With that said, only some $CO_2$ and $CH_4$ source sectors have obvious secondary tracers, and the associated atmospheric observations are primarily collected by in situ networks, not by satellites. Furthermore, progress in this area has been limited because of measurement availability, but this limitation could change in the future with more funding (i.e., in the case of radiocarbon) or deployment of new instrument technology (i.e., in the case of ethane).

## 5 Synthesis discussion

In this section, we synthesize progress to date on estimating sector-specific $CO_2$ and $CH_4$ emissions at state and national scale. We also discuss forward-looking opportunities to improve sector-specific GHG emissions estimates, with a particular focus on opportunities to integrate bottom-up and top-down strategies.

Recent innovations in both bottom-up and top-down efforts have advanced scientists' abilities to identify emissions from specific source sectors. Several efforts have produced high resolution, sector-specific inventory products that are based on more accurate, detailed activity data and EFs. These products have largely been driven by research in academia and by the Joint Research Centre in Europe. New inverse modeling strategies can incorporate these inventory estimates in more rigorous ways that are not limited to the spatial patterns in the inventory. In addition, more extensive observations are available to support these inverse modeling efforts, observations that span a number of spatial scales. For example, numerous intensive measurement campaigns in the past five years have focused on large GHG-emitting regions, particularly cities and oil and gas production basins. The national US in situ network and remote sensing GHG observations have also expanded in the last decade, though the US in situ network expansion is smaller than the level required for robust evaluation of a wide array of GHG source sectors.

Despite these advances in bottom-up inventories, top-down strategies, and measurement density, the scientific community has only been able to use inverse modeling and atmospheric data to improve sector-specific emissions estimates in a relatively small number of cases. To date, the community has had more success integrating top-down and bottom-up estimates for $CH_4$ than for $CO_2$; the atmospheric signal from biospheric $CO_2$ fluxes often obscures the signal from fossil fuel emissions, except in some urban environments. National $CH_4$ inventory estimates are often uncertain by a factor of 2–3 at the sector level while $CO_2$ inventories typically agree to within 5% (Fig. 1). Arguably, the community has been able to use top-down inverse modeling to improve these inventories when they arguably stood to benefit most.

Specifically, the community has been most successful with top-down, sector-specific attribution in two types of scenarios: intensive measurement campaigns paired with local-scale inverse modeling and opportunistic cases. In the former case, the community has put substantial resources into intensive, local-scale measurement campaigns for a few specific source sectors. Measurements from each affected locality or region provide a puzzle piece, and the community has begun to assemble a cohesive, national-scale picture by amalgamating these individual pieces. The community has employed this strategy in the case of $CH_4$ emissions from oil and gas operations (e.g., the SENEX, SONGNEX, Barnett Coordinated Campaign, etc.) and, to a lesser degree, in the case of urban $CO_2$ emissions (including recent measurement efforts in Los Angeles, Salt Lake City, Boston, and Oakland). These campaigns typically provide a snapshot of current emissions and would need to be repeated in the future to estimate how emissions vary over time.

Other cases of successful source attribution have been largely opportunistic. In certain cases, the community had the right atmospheric measurements and spatially-distinct source sectors to attribute emissions at large spatial scales. For example, Miller et al. (2013) found large $CH_4$ emissions in Texas and Oklahoma that did not fit the spatial distribution of cows, and $CH_4$ measurements in that region correlated with measurements of higher order alkanes. The authors concluded that a large fraction

of those emissions were likely due to oil and gas operations. A study using satellite observations from GOSAT reached similar conclusions (Turner et al., 2015).

Numerous future opportunities would improve scientists' ability to merge bottom-up inventories, inverse modeling, and atmospheric GHG data for better GHG source attribution:

*1. Combine the strengths of existing datasets*

The majority of inverse modeling studies to date have used only in situ or satellite GHG data to estimate emissions. $CH_4$ inverse modeling studies for North America provide a good example. Miller et al. (2013) used in situ observations from long term monitoring stations, Wecht et al. (2014a) used remote sensing observations from Envisat/SCIAMACHY, and Turner et al. (2015) used remote sensing observations from GOSAT. Future studies may be able to attribute emissions more effectively by leveraging the strengths of all available in situ and remote sensing datasets. Different datasets often bring complementary strengths for this attribution: remote sensing datasets have broad spatial coverage and in situ datasets have complete temporal coverage and greater sensitivity to surface emissions, among other strengths. A number of challenges may have prevented the synthesis of multiple datasets in past studies: large datasets entail a number of computational challenges, the data are not always accessible, and the observations can have different information content or error characteristics that are challenging to balance in a single framework. Future efforts that can combine these disparate datasets likely stand the best chance of attributing emissions to specific source sectors.

*2. Expand several existing measurement strategies*

Expanded GHG measurements would also advance efforts to attribute emissions to specific source sectors. As discussed earlier, some of the most successful top-down efforts to attribute emissions have been intensive aircraft campaigns. These campaigns are more flexible than the long term monitoring network and can easily target source sectors of interest by flying in specific regions, in flight patterns that encapsulate the source of interest, and by flying at certain times of year that have fewer competing biogenic sources. An expansion of these campaigns would enable scientists to target specific source sectors, including $CO_2$ emissions from large power plants, $CH_4$ from agriculture, and $CH_4$ from coal mines, among other source sectors. These aircraft campaigns could then be used to estimate regional-scale EFs. Existing aircraft campaigns, for example, have have estimated $CH_4$ leakage rates for a range of different oil and gas drilling basins (see Sects. 3.1 – 3.2). The long term in situ atmospheric network and GHG monitoring satellites could be used to intelligently extrapolate and gap-fill these regional EFs at larger spatial scales and to identify broad trends over time.

In addition, successful cases of sector-specific attribution have usually involved observations that span multiple spatial and temporal scales. This strategy allows scientists to bridge between the regional scale that atmospheric observations are best able to constrain and the facility-level scale where inventories are strongest. For example, atmospheric observations can be used to identify regional differences between top-down and bottom-up estimates. Subsequent facility-level and on-road measurements can indicate why those regional differences occurred and how to improve EFs in a way that will bring inventories into agreement with top-down estimates. This measurement strategy can be expensive and requires extensive coordination, but it has been used successfully in the case of oil and gas $CH_4$ emissions (e.g., Allen, 2014; Brandt et al., 2014; Peischl et al., 2015). Bottom-up and top-down estimates of these emissions disagree at regional and national spatial scales (e.g., Miller et al., 2013; Turner et al.,

2015). Subsequent facility and on-road measurements revealed that a small number of facilities account for a large percentage of emissions; EFs that account for this skewed distribution are more consistent with regional top-down estimates (e.g., Brantley et al., 2014; Lavoie et al., 2015; Subramanian et al., 2015).

Effective source attribution will also likely require the use of secondary tracers. Measurements of some secondary tracers, like ethane, have expanded markedly in the past several years with advances in instrumentation. With that said, measurements of tracers like radiocarbon are only available for some of the long term US monitoring sites.

*3. Improve inverse modeling strategies with an eye toward secondary tracers*

The inverse modeling community has yet to develop inverse modeling strategies that can fully leverage observations of secondary tracers. This task is not straightforward and would likely require the development of new strategies. These strategies would need to quantify heterogeneities in the ethane content of natural gas or the disequilibrium effect in the case of radiocarbon. Furthermore, these strategies would need to relate the primary and secondary tracers in a single statistical framework and would need to account for uncertainties in that relationship. Observations of these secondary tracers have historically been very sparse, so few studies have focused on designing statistical inverse modeling frameworks to fully exploit these tracers.

*4. Develop detailed activity data as part of bottom-up efforts*

Top-down efforts, like those outlined above, can help in developing regional-scale EFs for different source sectors. These studies can be particularly helpful when EFs are challenging to determine at facility scale. For example, direct measurements of oil and gas facilities are difficult to design because a small number of leaks account for the majority of emissions, and these large emitters may be difficult to find and/or representatively sample (see Sect. 2.3).

In contrast to EFs, activity data can only come from bottom-up inventory efforts. In fact, top-down efforts depend upon reliable activity data for attributing emissions (Sects. 3.1 and 3.3). Efforts to improve these activity datasets would markedly improve source attribution. In many cases, these activity data exist but are not publicly available or are not available in gridded form. Gurney et al. (2007) cite local fuel sales or electric utility bills as examples. $CH_4$ emissions from oil and gas provide an additional example. Oil and gas wells generally report production figures to state regulatory agencies, but this reporting varies by state, does not have a consistent format, and can be difficult to find (e.g., http://pmc.ucsc.edu/~brodsky/wellindex.html). The inaccessibility of accurate activity data for oil and gas operations has been a barrier to source attribution in recent national-scale $CH_4$ inverse modeling studies (Miller et al., 2013; Turner et al., 2015). Recent work by Maasakkers et al. (2016) created gridded versions of EPA's activity data and represents an important step forward. These activity data are key to connecting inverse modeling results with bottom-up estimates of specific source sectors. Future bottom-up efforts should particularly focus on the development and public release of gridded activity data.

In synthesis, future improvements in bottom-up inventories and top-down strategies would likely complement one another and translate into more reliable, sector-specific emissions estimates; scientists will likely need to combine both strategies to robustly estimate GHG emissions from individual sources. Improved activity data would lead to gridded inventory estimates with more accurate spatial and temporal patterns. Top-down frameworks could then harness these patterns, along with more extensive, future GHG observations, to estimate regional-scale EFs for specific source sectors. National-scale observations of secondary tracers like radiocarbon and ethane would further strengthen these top-down efforts for applicable source sectors.

This coordinated, combined approach offers the most promising opportunity to evaluate state and national GHG emissions reduction policies in the US.

*Acknowledgements.* We thank John B. Miller of NOAA, Thomas Nehrkorn of AER, Inc., Seongeun Jeong of Lawrence Berkeley Labs, and Yoichi Shiga of the Carnegie Institution for Science for their technical input on the manuscript. This work is funded by the Carnegie Distinguished Postdoctoral Fellowship.

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

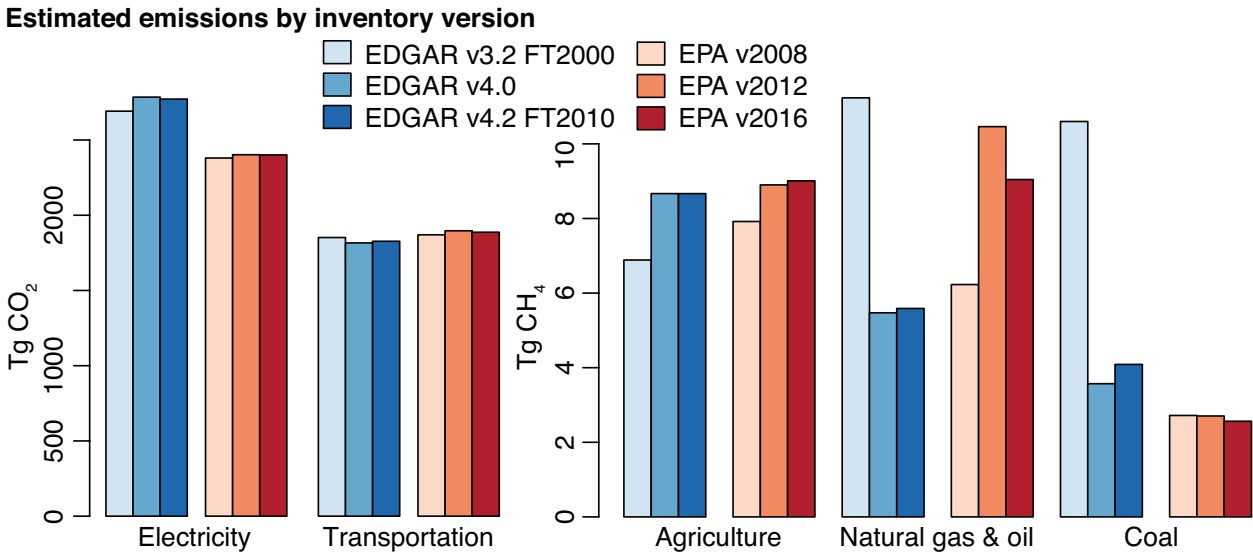

**Figure 1.** EDGAR and EPA inventory emissions estimates for different US fossil fuel source sectors (Olivier et al., 2014; EPA, 2016a), including several versions of each inventory. $CO_2$ estimates are consistent between EPA and EDGAR and among inventory versions. $CH_4$ estimates, however, vary widely, an indication of uncertainty in $CH_4$ emissions. All of the estimates are for 2005 except for EDGAR FT2000 which is for 2000. Note that EDGAR includes $CO_2$ from heating in its electricity estimate while EPA does not. As a result, the EDGAR $CO_2$ estimate is higher than EPA's estimate.

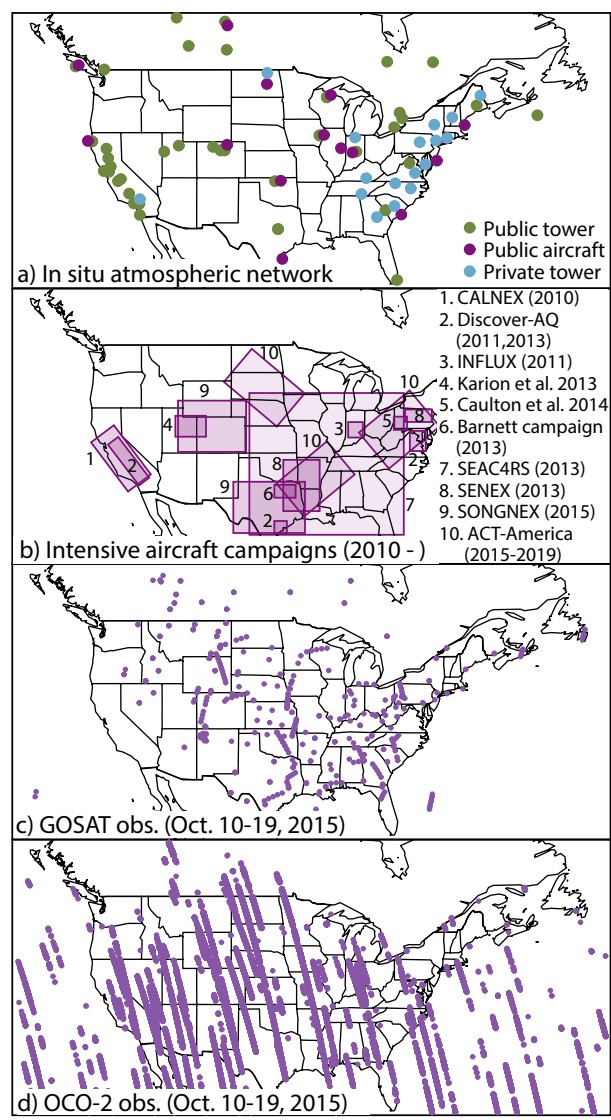

**Figure 2.** This figure highlights different $CO_2$ observation networks and how the spatial coverage of those networks differ. These networks include tower and regular aircraft sampling sites (**a**); several recent, intensive aircraft campaigns (**b**); the GOSAT satellite (**c**); and the OCO-2 satellite (**d**). Note that the dots on each panel are not equivalent; an in situ monitoring sites in panel **a** often provides continuous or daily data while each dot in panels **c** (GOSAT) and **d** (OCO-2) indicates the location of a single observation. Public towers and public aircraft sites are operated by NOAA, DOE, Environment Canada, and partners, and the sites shown are current through 2016. Private towers are operated by Earth Networks, and the locations here are current through 2012. Most tower and aircraft sites also include $CH_4$ observations.

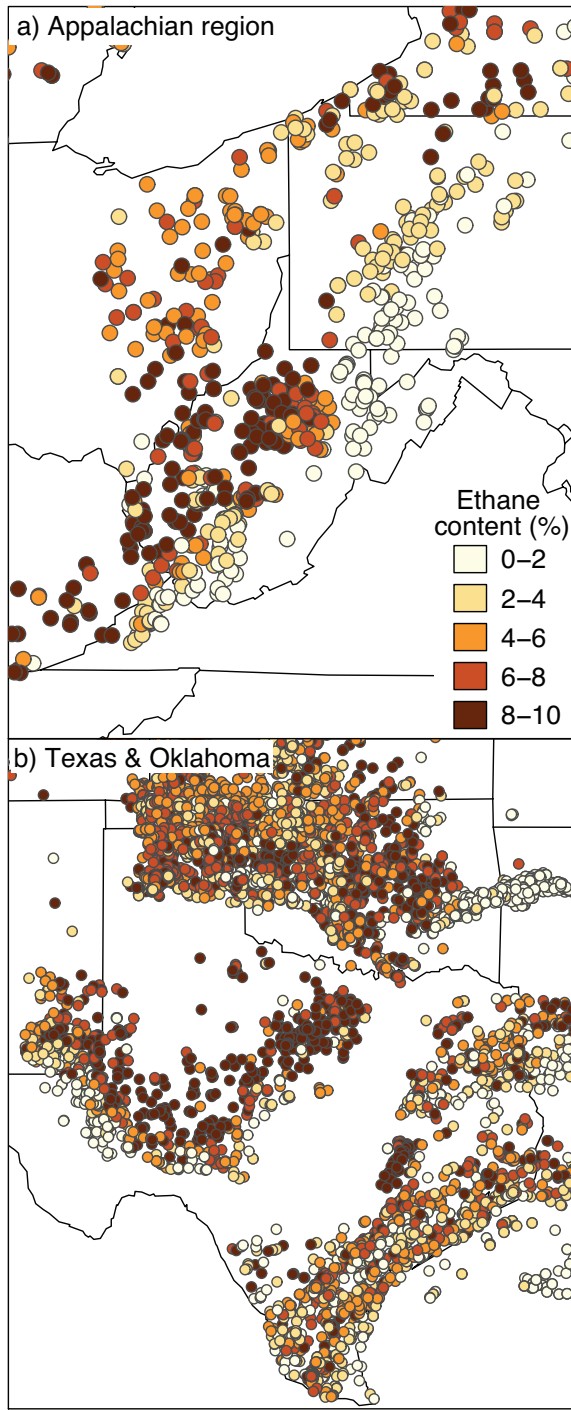

**Figure 3.** This figure shows the variability in ethane content of natural gas for two major drilling regions of the United States. Ethane content is a key parameter when estimating oil and gas $CH_4$ emissions using atmospheric ethane measurements. The samples show substantial heterogeneity in some regions (e.g., Oklahoma) and exhibit clear spatial patterns in other regions (e.g., Texas and West Virginia). All data in this figure are from the USGS Geochemistry Laboratory Database (USGS Energy Resources Program, 2015).