# Peer review of "Constraining sector-specific CO2 and CH4 emissions in the United States"

_Atmospheric Chemistry and Physics, 2016_

## Referee Comment (RC1) · Anonymous Referee #1 · 31 Aug 2016

This paper reviews methods and datasets for estimating or constraining sector specific emissions of CO2 and CH4 in the US. The review paper refers to quite recent publications. The paper is well written, and I recommend publishing after the following minor comments are addressed.

Specific comments: In the introduction there should be some mentioning of INDCs (Intended Nationally Determined Contributions), which were decided during COP 21 in Paris 2015.

P3 L1-2: I suggest reformulating to "frameworks that can synergistically leverage the information content of bottom-up datasets and top-down strategies using atmospheric GHG data"

[Figure]

P3 L7: May be reformulate "to attribute that trend to a specific source sector(s)" to e.g. "to attribute this trend to trends in specific source sectors"

P4 L27: A reference for EDGAR needs to be included here.

P15 L22: I think a reference to Dils et al., 2014, which systematically validates CH4 and CO2 products from GOSAT against TCCON data, would be appropriate: Dils, B., Buchwitz, M., Reuter, M., Schneising, O., Boesch, H., Parker, R., Guerlet, S., Aben, I., Blumenstock, T., Burrows, J. P., Butz, A., Deutscher, N. M., Frankenberg, C., Hase, F., Hasekamp, O. P., Heymann, J., De Mazière, M., Notholt, J., Sussmann, R., Warneke, T., Griffith, D., Sherlock, V. and Wunch, D.: The Greenhouse Gas Climate Change Initiative (GHG-CCI): comparative validation of GHG-CCI SCIAMACHY/ENVISAT and TANSO-FTS/GOSAT CO2 and CH4 retrieval algorithm products with measurements from the TCCON, Atmos. Meas. Tech., 7(6), 1723–1744, doi:10.5194/amt-7-1723-2014, 2014.

P15 L28: Here I think the CarbonSat mission should be mentioned, as it combines high spatial resolution with a large swath, making it useful for emission detection. Some relevant papers are listed here: Buchwitz, M., Reuter, M., Bovensmann, H., Pillai, D., Heymann, J., Schneising, O., Rozanov, V., Krings, T., Burrows, J. P., Boesch, H., Gerbig, C., Meijer, Y. and Löscher, A.: Carbon Monitoring Satellite (CarbonSat): assessment of atmospheric CO2 and CH4 retrieval errors by error parameterization, Atmos. Meas. Tech., 6(12), 3477–3500, doi:10.5194/amt-6-3477-2013, 2013. Pillai, D., Buchwitz, M., Gerbig, C. and Koch, T.: Tracking city CO2 emissions from space using a high resolution inverse modeling approach: A case study for Berlin, Germany, Atmos. Chem. Phys., doi:10.5194/acp-16-9591-2016, 2016.

P17 L17: reword "now markets and ethane analyzer" -> "now markets an ethane analyzer"

---

## Short Comment (SC1) · 8 Nov 2016

Hi Scot,

Nice to see your manuscript on this topic – very useful.

I just have a quick comment. Here are two recent papers that also investigate spatial and temporal limits on emissions constraints from current and future remote sensing instruments, which you might consider citing in your work (e.g., section 4.1):

Bousserez, N., Henze, D. K., Rooney, B., Perkins, A., Wecht, K. J., Turner, A. J., Natraj, V., and Worden, J. R.: Constraints on methane emissions in North America from future geostationary remote-sensing measurements, Atmos. Chem. Phys., 16, 6175-6190,

doi:10.5194/acp-16-6175-2016, 2016.

Jacob, D. J., Turner, A. J., Maasakkers, J. D., Sheng, J., Sun, K., Liu, X., Chance, K., Aben, I., McKeever, J., and Frankenberg, C.: Satellite observations of atmospheric methane and their value for quantifying methane emissions, Atmos. Chem. Phys. Discuss., doi:10.5194/acp-2016-555, in review, 2016.

Best,

Daven
* * *

---

## Referee Comment (RC2) · Anonymous Referee #2 · 10 Nov 2016

General Comments:

This is a helpful summary of recent work to estimate methane and CO2 emissions from the US and should be published with minor revisions. The scope of the review should be stated in the introduction. For example, many studies aimed at understanding CO2 uptake by terrestrial vegetation are evidently out of scope, even though biological CO2 sequestration may significantly offset US CO2 emissions.

Discussion of the quality of satellite data required for anthropogenic flux estimation and trend estimation would be helpful. The measurement requirements to detect anthropogenic CO2 plumes are described in the 2010 NRC Report, Verifying Greenhouse Gas Emissions, and in publications de-

scribing the notional CarbonSat mission as well as in the CarbonSat report (http://esamultimedia.esa.int/docs/EarthObservation/SP1330-1_CarbonSat.pdf).

Also it should be noted that the current generation of satellite sensors are not designed to provide comprehensive global mapping and are therefore not ideally suited for urban and point/source estimation. OCO-2 and GOSAT were designed for global carbon cycle science rather than emissions monitoring. How does the uncertainty in e.g. the Kort et al. analysis of Los Angeles emissions using GOSAT compare with the requirements for useful urban trend detection (e.g., something like a 10% reduction in emissions over 10 years) ?

It would be useful to see more discussion about where existing inventories and/or inversions agree and where they disagree. For example, how do the Schneising et al SCIAMACHY fugitive methane emissions estimates for North America compare with those from aircraft campaigns?

Finally, some more discussion of transport modeling errors would be useful. To what extent do uncertainties in simulated transport limit top-down flux estimation? What type of work is needed to address transport uncertainty?

Specific Comments:

page 2, line 20: Are there any regulations targeting CH4 emissions from agriculture? Perhaps worth mentioning here that agriculture is a large source of CH4 even if not regulated yet.

page 3, line 9: "meteorically" sounds sensational

page 3, line 15: Perhaps briefly discuss biological CO2 sinks and potential for deliberate sequestration, along with concomitant need for verification of such reservoirs. Also could mention challenges of accounting for emissions from CH4 wetlands, as well as CH4 emissions related to anthropogenic interference in the hydrological systems (emissions from reservoirs). Something about co-location of cows and oil and gas

perhaps also worth mentioning here.

page 4, line 1: I don't see a reference for EDGAR inventory in this list of references for global efforts, though it is frequently used.

page 4, line 10: For the example of coal gasification, how is energy lost in conversion of coal to gas taken into account? It seems like this should count as emissions from coal.

page 4, line 28: First mention of EDGAR, but I don't see any reference. Perhaps add a url.

page 5, first paragraph: Perhaps mention for which years these products are available and how often they are updated (or not updated).

page 5 line 8: instead of "rigorous" consider "detailed"

page 5 line 14: repeated use of "EFs" results in confusing long sentence. Consider simplifying e.g., "…much higher EFs that result in higher emissions that are much more consistent…"

page 5, various lines: over-use of the word "leverage" in this section

page 5, line 26: The Andres et al. effort is also government-sponsored

page 5, line 31: "these omissions" since threshold plus ag exemption

page 6, line 20: it would be helpful to define what is meant by on-road measurements, i.e. are these all ground-based mobile using public (or private) roads?

page 6, line 25: Smokestack measurements of CO2 are not used in the EPA inventory?

page 7, line 2: Marcellus not Mercellus

page 7, line 20: Could you also include agricultural CH4 emissions in Figure 1?

page 9, line 10: Mays and Cambaliza both Indianapolis.

page 9, paragraph beginning on line 31: A limitation is that most of these studies use data from a single campaign and provide only a snapshot of emissions. Some of the studies used tracers such as ethane to estimate contribution of landfills, etc. I think this is worth mentioning here.

page 11, line 33. The verbiage "run an atmospheric transport model once per source sector" is confusing. Zhao et al. and Jeong et al. used STILT-WRF, so they generated footprints from a single WRF run. Suggest simply eliminating the phrase "once per source sector", since details of how the transport model is run may vary.

equation 5: x[i] not defined.

page 12, line 28: Technicality: SCIAMACHY is not a satellite. It is the name of a sensor on the Envisat satellite.

page 13, equation 6: A limitation of the GIM as implemented in the cited references is that the betas are spatially constant whereas in reality relationships between activity data and emissions may vary spatially or temporally.

page 14, line 10: Radiocarbon measurements show that respired biogenic CO2 is significant even in winter.

page 15, line 15: It should be mentioned that in order for satellite measurements to be useful for understanding and tracking urban emissions, they must not only detect the presence of a large urban area but also be sufficiently sensitive to measure trends.

page 15, line 24: Limitations of the ASCENDS concept should be mentioned. For example ASCENDS will provide limited spatial coverage, infrequent revisits, and will low signal to noise for urban signatures.

page 15, line 30: Revised launch date needed for TROPOMI.

page 16, line 11: More recently than what?

page 16, line 22: The description of current radiocarbon sampling could be improved.

More specificity is needed, especially regarding the temporal density of samples in the current network compared to what is recommended by Basu et al.

page 17, line 2: Impact of disequilibrium fluxes on estimated emissions can be mitigated if major urban areas have both upwind and downwind sampling.

page 17, line 17: Typo "now markets and".

page 17, line 18: Detlev Helmig's lab at INSTAAR has been measuring ethane in whole air samples from the NOAA global network for many years (http://www.nature.com/ngeo/journal/v9/n7/abs/ngeo2721.html). There is a also a new instrument that is now being used to measure ethane from whole air samples North American tall towers and aircraft.

page 18, lines 15-20: Repeated use of "far more". Not quantitative.

page 18, line 25: Perhaps should point out that intensive measurement campaigns provide only a snapshot and, unless repeated, provide no information about how emissions may vary over time.

page 19, line 13: I don't think it is helpful or fair to single out Environment Canada for criticism (especially since focus of this review is US emissions), though your general point about data not being readily accessible is valid. CO2 data from Environment Canada through 2015 is available from the GLOBALVIEWplus_v2.1 ObsPack available here (http://www.esrl.noaa.gov/gmd/ccgg/obspack/data.php). Hopefully a similar product will be available soon for CH4.
* * *

---

## Author Comment (AC1) · 29 Dec 2016

Thank your for the suggested references; these are very helpful. We have added them in to the revised version of the manuscript.
* * *

---

## Author Comment (AC2) · 29 Dec 2016

Thank you for the ideas and suggestions for the review paper. They have greatly helped us improve the manuscript. Below, we have listed each of the suggestions and the corresponding revisions that we have made to the manuscript.

- In the introduction there should be some mentioning of INDCs (Intended Nationally Determined Contributions), which were decided during COP 21 in Paris 2015.

  This is a great suggestion. We have included INDCs in the revised introduction.

- P3 L1-2: I suggest reformulating to "frameworks that can synergistically leverage the information content of bottom-up datasets and top-down strategies using atmospheric GHG data"

  We have updated this sentence accordingly.

- P3 L7: May be reformulate "to attribute that trend to a specific source sector(s)" to e.g. "to attribute this trend to trends in specific source sectors"

  We have revised the sentence accordingly. The new wording sounds more precise.

- P4 L27: A reference for EDGAR needs to be included here.

  We have added a reference to EDGAR in this line.

- P15 L22: I think a reference to Dils et al., 2014, which systematically validates CH4 and CO2 products from GOSAT against TCCON data, would be appropriate: Dils, B., Buchwitz, M., Reuter, M., Schneising, O., Boesch, H., Parker, R., Guerlet, S., Aben, I., Blumenstock, T., Burrows, J. P., Butz, A., Deutscher, N. M., Frankenberg, C., Hase, F., Hasekamp, O. P., Heymann, J., De Mazière, M., Notholt, J., Sussmann, R., Warneke, T., Griffith, D., Sherlock, V. and Wunch, D.: The Greenhouse Gas Climate Change Initiative (GHG-CCI): comparative validation of GHG-CCI SCIAMACHY/ENVISAT and TANSO-FTS/GOSAT CO2 and CH4 retrieval algorithm products with measurements from the TCCON, Atmos. Meas. Tech., 7(6), 1723–1744, doi:10.5194/amt-7-1723- 2014, 2014.

  This is a great suggestion. We have added this reference to the corresponding line of the revised manuscript.

- P15 L28: Here I think the CarbonSat mission should be mentioned, as it combines high spatial resolution with a large swath, making it useful for emission detection. Some rel- evant papers are listed here: Buchwitz, M., Reuter, M.,

Bovensmann, H., Pillai, D., Hey- mann, J., Schneising, O., Rozanov, V., Krings, T., Burrows, J. P., Boesch, H., Gerbig, C., Meijer, Y. and Löscher, A.: Carbon Monitoring Satellite (CarbonSat): assessment of atmospheric CO2 and CH4 retrieval errors by error parameterization, Atmos. Meas. Tech., 6(12), 3477–3500, doi:10.5194/amt-6-3477-2013, 2013. Pillai, D., Buchwitz, M., Gerbig, C. and Koch, T.: Tracking city CO2 emissions from space using a high reso- lution inverse modeling approach: A case study for Berlin, Germany, Atmos. Chem. Phys., doi:10.5194/acp-16-9591-2016, 2016.

CarbonSat was a notable shortfall in the initial manuscript. We have added several lines on CarbonSat and GeoCARB to this section (along with the references above). Thank you for including these suggested references; they are very helpful.

- P17 L17: reword "now markets and ethane analyzer" -> "now markets an ethane analyzer"

Thank you for pointing out this typo. We have fixed it in the revised manuscript.

---

## Author Comment (AC3) · 29 Dec 2016

Thank you for the thorough and highly constructive suggestions on the manuscript. The suggestions are insightful and thoughtful and have been incredibly helpful for improving the manuscript.

- The scope of the review should be stated in the introduction. For example, many studies aimed at understanding CO2 uptake by terrestrial vegetation are evidently out of scope, even though biological CO2 sequestration may significantly offset US CO2 emissions.

  This is a great suggestion for clarifying the manuscript framing. We have added

content to the introduction defining the scope as suggested here. The reviewer makes a great point about biological $CO_2$ sequestration. We felt that this topic would have expanded the scope of the review beyond what we could feasibly cover in a single paper. It would be an excellent topic for a future review paper, though.

• Discussion of the quality of satellite data required for anthropogenic flux estimation and trend estimation would be helpful. The measurement requirements to detect anthropogenic CO2 plumes are described in the 2010 NRC Report, Verifying Greenhouse Gas Emissions, and in publications describing the notional CarbonSat mission as well as in the CarbonSat report (http://esamultimedia.esa.int/docs/EarthObservation/SP1330-1_CarbonSat.pdf).

We have added this information to Sect. 4.1 of the revised manuscript. In addition, we have also added mention of the newly announced GeoCARB satellite in this section.

• Also it should be noted that the current generation of satellite sensors are not designed to provide comprehensive global mapping and are therefore not ideally suited for urban and point/source estimation. OCO-2 and GOSAT were designed for global carbon cycle science rather than emissions monitoring. How does the uncertainty in e.g. the Kort et al. analysis of Los Angeles emissions using GOSAT compare with the requirements for useful urban trend detection (e.g., something like a 10

The reviewer makes a great point, and we have added this information to Sect. 4.1. Current satellite data products are unlikely to detect a 10% trend over 10 years; the measurement uncertainties and retrieval biases associated with these products are likely too large relative to the $XCO_2$ increment. Kort et al. (2012) estimate that GOSAT could detect a trend as small as 22% from Los Angeles. However, Los Angeles is arguably an ideal case study, a very large city with a

small biospheric $CO_2$ signal.

- It would be useful to see more discussion about where existing inventories and/or inversions agree and where they disagree. For example, how do the Schneising et al SCIAMACHY fugitive methane emissions estimates for North America compare with those from aircraft campaigns?

We have added text to Sect. 2.4 that highlights where existing estimates agree and disagree. Different top-down and inventory studies often have very different spatial scales and cover different time windows. These differences in scale can make disagreements among the estimates more challenging to identify. We also highlight this point in the revised version of Sect. 2.4.

- Finally, some more discussion of transport modeling errors would be useful. To what extent do uncertainties in simulated transport limit top-down flux estimation? What type of work is needed to address transport uncertainty?

We have added content to the synthesis discussion in Sect. 5. We mention the importance of reducing transport errors and recent innovations that could aid in this effort (e.g., monitoring mixed layer height with LIDAR). In sections 3.1 and 3.3, we also highlight studies that discuss the impact of transport errors on sector-specific attribution. These studies include papers by Shiga et al. (2014) and Karion et al. (2015).

- page 2, line 20: Are there any regulations targeting CH4 emissions from agriculture? Perhaps worth mentioning here that agriculture is a large source of CH4 even if not regulated yet.

To our knowledge, there are no regulations that mandate $CH_4$ emissions reductions from agriculture in the U.S.. In August of 2014, the US EPA, USDA, and US DOE released the "Biogas Opportunities Roadmap" targeting voluntary reduction strategies for agriculture

(https://www3.epa.gov/climatechange/Downloads/Biogas-Roadmap.pdf). We have updated this line of the manuscript with a brief mention of the roadmap.

- page 3, line 9: "meteorically" sounds sensational

  We have replaced this phrase with "began in the past decade."

- page 3, line 15: Perhaps briefly discuss biological CO2 sinks and potential for deliberate sequestration, along with concomitant need for verification of such reservoirs. Also could mention challenges of accounting for emissions from CH4 wetlands, as well as CH4 emissions related to anthropogenic interference in the hydrological systems (emissions from reservoirs). Something about co-location of cows and oil and gas perhaps also worth mentioning here.

  We have added a sentence to this paragraph mentioning the possibility of biological or geological sequestration as a public policy tool and the need to verify those carbon sinks.

- page 4, line 1: I don't see a reference for EDGAR inventory in this list of references for global efforts, though it is frequently used.

  We have added the following reference to EDGAR: European Commission, Joint Research Centre (JRC)/Netherlands Environmental Assessment Agency (PBL). Emission Database for Global Atmospheric Research (EDGAR), release version 4.3.1 http://edgar.jrc.ec.europa.eu/overview.php?v=431, 2016.

  This particular publication is one of the preferred references for EDGAR stated on their web site (http://edgar.jrc.ec.europa.eu/terms_of_use.php).

- page 4, line 10: For the example of coal gasification, how is energy lost in conversion of coal to gas taken into account? It seems like this should count as emissions from coal.

  We have clarified this line in the manuscript. This line does not refer to energy lost in the conversion of coal to gas. Rather, the energy converted from coal to

gas is moved from the "industrial other coal" category in EPA's accounting to the "natural gas combustion" category. EPA explains, "The energy in this synthetic natural gas enters the natural gas distribution stream, and is accounted for in EIA natural gas combustion statistics. Because this energy of the synthetic natural gas is already accounted for as natural gas combustion, this amount of energy is deducted from the industrial coal consumption statistics to avoid double counting" (EPA 2016c, p. A-31).

- page 4, line 28: First mention of EDGAR, but I don't see any reference. Perhaps add a url.

  We have added a reference to EDGAR (JRC/PBL 2016, as shown above).

- page 5, first paragraph: Perhaps mention for which years these products are available and how often they are updated (or not updated).

  We have added this information to the paragraph.

- page 5 line 8: instead of "rigorous" consider "detailed"

  We have replaced the words as suggested.

- page 5 line 14: repeated use of "EFs" results in confusing long sentence. Consider simplifying e.g., "...much higher EFs that result in higher emissions that are much more consistent. . ."

  We have simplified and shortened this sentence accordingly.

- page 5, various lines: over-use of the word "leverage" in this section

  We have reduced the usage of this word throughout this section.

- page 5, line 26: The Andres et al. effort is also government-sponsored

[Figure]

This statement is technically true since Andres works at Oak Ridge National Laboratory. What we intended to say is that most of these inventories are not constructed by regulatory agencies and are not part of an official regulatory agency inventory product. We have revised this line accordingly.

- page 5, line 31: "these omissions" since threshold plus ag exemption

  We have changed "this" to "these" as suggested.

- page 6, line 20: it would be helpful to define what is meant by on-road measurements, i.e. are these all ground-based mobile using public (or private) roads?

  All of the studies listed here use ground-based mobile measurements on roadways. Not all of these studies list whether the roads were public or private (e.g., Mitchell et al., 2015, Subramanian et al., 2015). With that said, many of the studies listed in the manuscript use public roadways (e.g., Brondfield et al., 2012; Brantley et al., 2014; Jackson et al., 2014; Lan et al., 2015; Maness et al., 2015; and Roscioli et al., 2015). Roscioli et al., (2015) also took ground-based mobile measurements within the perimeter of many sites.

- page 6, line 25: Smokestack measurements of CO2 are not used in the EPA inventory?

  We have edited this passage to make it more precise. EPA uses smokestack measurements in some contexts. For example, smokestack or facility level measurements are used in EPA's Greenhouse Gas Reporting Program (GHGRP) (e.g., see Sect. 2.1 in https://www.epa.gov/sites/production/files/2016-03/documents/stationaryemissions_3_2016.pdf).

- page 7, line 2: Marcellus not Mercellus

  We have corrected this spelling accordingly.

- page 7, line 20: Could you also include agricultural CH4 emissions in Figure 1?

We have added agricultural emissions to the figure.

- page 9, line 10: Mays and Cambaliza both Indianapolis.

  This is correct. One study examined $CO_2$ while the other focused on $CH_4$ (and hence we have listed the studies separately).

- page 9, paragraph beginning on line 31: A limitation is that most of these studies use data from a single campaign and provide only a snapshot of emissions. Some of the studies used tracers such as ethane to estimate contribution of landfills, etc. I think this is worth mentioning here.

  We have added this point to the paragraph in question.

- page 11, line 33. The verbiage "run an atmospheric transport model once per source sector" is confusing. Zhao et al. and Jeong et al. used STILT-WRF, so they generated footprints from a single WRF run. Suggest simply eliminating the phrase "once per source sector", since details of how the transport model is run may vary.

  We have eliminated this phrase accordingly in the revised manuscript.

- equation 5: x[i] not defined.

  The variable x[i] is defined in the lines of text following Eq. 3. We have added a clarifying note following Eq. 5: "All other variables are as defined earlier."

- page 12, line 28: Technicality: SCIAMACHY is not a satellite. It is the name of a sensor on the Envisat satellite.

  We have updated this line in the text accordingly.

- page 13, equation 6: A limitation of the GIM as implemented in the cited references is that the betas are spatially constant whereas in reality relationships between activity data and emissions may vary spatially or temporally.

[Figure]

This is true, and we have added this point to the revised manuscript. The coefficients could, in theory, be variable. For example, Fang and Michalak (2015, doi:10.1002/2014GB005034) allow the coefficients to vary from one biome to another (in the context of biospheric fluxes). Gelfand et al. (2003, doi:10.1198/016214503000170) developed a statistical model that estimates spatially variable coefficients, albeit not in the context of atmospheric inverse modeling. Either of these studies could be a starting point for implementing variable coefficients within a GIM.

- page 14, line 10: Radiocarbon measurements show that respired biogenic CO2 is significant even in winter.

  This is definitely true. However, the overall magnitude and diurnal variability of biogenic $CO_2$ fluxes are much lower in the winter than in summer. As a result, inverse modeling approaches stand a better chance of identifying fossil fuel flux patterns in winter than in summer (e.g., Shiga et al. 2014).

- page 15, line 15: It should be mentioned that in order for satellite measurements to be useful for understanding and tracking urban emissions, they must not only detect the presence of a large urban area but also be sufficiently sensitive to measure trends.

  This is an astute point, and we have added discussion on this point to the corresponding lines of the revised manuscript. For example, Hammerling et al. (2015) examined whether a LIDAR mission like ASCENDS would be able to detect changes in anthropogenic emissions from large regions like Europe or China.

- page 15, line 24: Limitations of the ASCENDS concept should be mentioned. For example ASCENDS will provide limited spatial coverage, infrequent revisits, and will low signal to noise for urban signatures.

  Good suggestion. We agree that this information is important to mention and have added it to the corresponding lines of the revised manuscript.

- page 15, line 30: Revised launch date needed for TROPOMI.

  We have revised the launch date. The TROPOMI team currently estimates a launch date sometime in 2017 (http://www.tropomi.eu/instrument/status-0).

- page 16, line 11: More recently than what?

  We have replaced the phrase "more recently" with "Beginning in the 1940s, . . .."

- page 16, line 22: The description of current radiocarbon sampling could be improved. More specificity is needed, especially regarding the temporal density of samples in the current network compared to what is recommended by Basu et al.

  We have added more specific information on the available radiocarbon observations at tall tower and aircraft sites in the US.

- page 17, line 2: Impact of disequilibrium fluxes on estimated emissions can be mitigated if major urban areas have both upwind and downwind sampling.

  Good point. We have added this information into the corresponding lines of the revised manuscript.

- page 17, line 17: Typo "now markets and".

  We have fixed this typo in the revised manuscript.

- page 17, line 18: Detlev Helmig's lab at INSTAAR has been measuring ethane in whole air samples from the NOAA global network for many years (http://www.nature.com/ngeo/journal/v9/n7/abs/ngeo2721.html). There is a also a new instrument that is now being used to measure ethane from whole air samples North American tall towers and aircraft.

  We have updated these lines of the revised manuscript to reflect this information.

- page 18, lines 15-20: Repeated use of "far more". Not quantitative.

  We have replaced the words "far more" in the corresponding paragraph.

- page 18, line 25: Perhaps should point out that intensive measurement campaigns provide only a snapshot and, unless repeated, provide no information about how emissions may vary over time.

  This is a great point, and we have added it to the corresponding lines of the revised manuscript.

- page 19, line 13: I don't think it is helpful or fair to single out Environment Canada for criticism (especially since focus of this review is US emissions), though your general point about data not being readily accessible is valid. CO2 data from Environment Canada through 2015 is available from the GLOBALVIEWplus_v2.1 ObsPack available here (http://www.esrl.noaa.gov/gmd/ccgg/obspack/data.php). Hopefully a similar product will be available soon for CH4.

  We have removed this reference from the revised manuscript.

---

## Author Response (AR2)

To the editor:

Thank you for your suggestions on the manuscript. These are extremely helpful. We have corrected these points in the manuscript and have submitted an updated version of the manuscript.

All the best,
Scot